

# Assessing the performance and explainability of an avalanche danger forecast model

Cristina Pérez-Guillén[1], Frank Techel[1], Michele Volpi[2], and Alec van Herwijnen[1]

[1]WSL Institute for Snow and Avalanche Research SLF, Davos, Switzerland*
[2]Swiss Data Science Center, ETH Zurich and EPFL, Switzerland

**Correspondence:** Cristina Pérez Guillén (cristina.perez@slf.ch)

**Abstract.**

During winter, public avalanche forecasts provide crucial information for professional decision-makers as well as recreational backcountry users in the Swiss Alps. While avalanche forecasting has traditionally relied exclusively on human expertise, the Swiss avalanche warning service has recently integrated machine-learning models to support the forecasting process.

This study assesses a random forest classifier trained with weather data and physical snow-cover simulations as input for predicting dry-snow avalanche danger levels during the initial live-testing in the winter season of 2020-2021. The model achieved ∼70% agreement with published danger levels, performing equally well in *nowcast-* and *forecast*-mode. By using model-predicted probabilities, continuous expected danger values were computed, showing a high correlation with the sub-levels as published in the Swiss forecast. The model effectively captured temporal dynamics and variations across different slope aspects and elevations, though it decreased the performance during periods with persistent weak layers in the snowpack. SHapley

Additive exPlanations (SHAP) were employed to make the model's decision process more transparent, reducing its 'black-box' nature. Beyond increasing the explainability of model predictions, the model encapsulates twenty years of forecasters' experience in aligning weather and snowpack conditions with danger levels. Therefore, the presented approach and visualization could also be employed as a training tool for new forecasters, highlighting relevant parameters and thresholds. In summary,

machine-learning models as the danger-level model, often considered 'black-box' models, can provide high-resolution, comparably transparent "second opinions" that complement human forecasters' danger assessments.

## 1 Introduction

The destructive potential of snow avalanches demands reliable forecasts and timely warnings to ensure safety and mobility in avalanche-prone regions. In Switzerland, snow avalanches are the deadliest natural hazard (Badoux et al., 2016), causing

significant economic losses that can exceed several hundred million Swiss francs (Bründl et al., 2004). Consequently, the Swiss avalanche warning service issues a daily avalanche bulletin during winter. This forecast provides essential information for the public and local authorities, aiding in decision-making for safe winter backcountry recreation and ensuring public safety on transportation routes and in settlements during periods of high avalanche danger.



Until now, public avalanche forecasting has been entirely a human-expert decision-making process during which avalanche
forecasters interpret a wide range of data sources such as weather forecasts, meteorological observations, and snow-cover
model outputs, combined with field observations of snow-cover stability and avalanche activity (Techel et al., 2022b). In fore-
cast products, the severity of expected avalanche conditions is described with a danger level for a region and specific time
period summarizing snowpack stability, frequency distribution, and avalanche size (Techel et al., 2020a; EAWS, 2021a). Ad-
ditionally, typical avalanche situations are also communicated using so-called "avalanche problems" (EAWS, 2021b). Despite
numerous attempts to employ statistical or machine-learning methods as alternatives to conventional avalanche forecasting
(e.g. Schweizer et al., 1994; Schirmer et al., 2009; Pozdnoukhov et al., 2011; Mitterer and Schweizer, 2013), very few of these
approaches were successfully integrated into operational avalanche forecasting.

In recent years, the growth in data volumes and advancements in data-driven modelling have opened up new possibilities for
developing avalanche forecasting models. Among them, different machine-learning models have emerged including models
predicting danger levels (Pérez-Guillén et al., 2022; Fromm and Schönberger, 2022), transferring these predictions to a format
similar to human-made forecasts (Maissen et al., 2024), identifying typical avalanche problem types (Horton et al., 2020;
Reuter et al., 2022), assessing snowpack stability and critical snow layers (Mayer et al., 2022; Herla et al., 2023), or predicting
avalanche activity (Dkengne Sielenou et al., 2021; Viallon-Galinier et al.; Hendrick et al., 2023; Mayer et al., 2023). These data-
driven approaches can provide relevant outputs for avalanche forecasting with high spatiotemporal resolution (van Herwijnen
et al., 2023). Despite rigorous model validation, which is indispensable for their integration into the forecasting process,
the 'black-box' nature of many machine learning models presents an additional challenge to their use. While predictions
generated by simpler models can be readily interpreted (Horton et al., 2020; Reuter et al., 2022; Mayer et al., 2023), the
'decision-making' behind complex machine-learning models generally remains hidden, making it challenging to understand
how variables are weighted to make predictions. Though various methods for explaining machine-learning predictions have
been proposed (Ribeiro et al., 2016; Lundberg and Lee, 2017, e.g.), so far, these have not been tested for models in an avalanche-
forecasting context.

With this study, we have two objectives: to evaluate the performance and to deepen the understanding of the predictions
generated by a model predicting the avalanche danger level (Pérez-Guillén et al., 2022). To this end, we analysed the live-
predictions during the initial winter season when the model was tested before avalanche forecasters started to integrate these
model predictions into their daily forecasting process. During this forecasting season, the model provided *nowcast* and 24-hour
*forecast* predictions for dry-snow conditions in Switzerland. Here, we compare aspect-specific predictions with danger level
forecasts in the public bulletin, including a comparison of model-predicted probabilities with sub-level qualifiers assigned to
danger levels (e.g. Techel et al., 2022b; Lucas et al., 2023). Furthermore, we assessed model performance in situations when
different avalanche problems were relevant. Finally, to understand the features driving predictions towards a certain danger
level, and thus to reduce the 'black-box' nature of the model, we employed an explanation model quantifying the impact of the
model's input features for each danger level, overall but also for individual predictions.



## 2 Public avalanche forecast in Switzerland

The avalanche forecast is published at 17:00 LT (local time), valid until 17:00 LT the following day, with an update at 08:00 LT, if necessary (SLF, 2023). The forecast domain is split into 149 warning regions (status 2021, average size $\approx 200 \ km^2$; white polygons in Fig. 1), which are clustered into danger regions based on the expected avalanche conditions (black polygons in Fig. 1). Each danger region is assigned a danger level based on the European Avalanche Danger Scale (EAWS, 2021a, levels: 1–Low, 2–Moderate, 3–Considerable, 4–High and 5–Very High)[1] summarizing the severity of avalanche conditions. Moreover, the locations of potential avalanche-triggering spots are specified by providing the aspects and elevation where these are most frequent (SLF, 2023). Within the specified aspect-elevation zone, the danger level is applicable (Fig. 1). In addition, the avalanche situation is communicated using the concept of the *typical avalanche problems* (EAWS, 2021b). For dry-snow conditions, three avalanche problems are used (SLF, 2023):

– The *New snow* problem (NS) indicates that the current or most recent snowfall can be released.

– The *Wind slab* problem (WS) is given when fresh or recently wind-drifted snow is of concern.

– The *Old snow* problem (OS) highlights that persistent weak layer(s) within the snowpack are potentially prone to triggering.

In the bulletin, only the avalanche problems that significantly contribute to the danger are mentioned. For dry-snow conditions, at most, two avalanche problems can be specified. If no particular avalanche problem is identified (e.g., at 1-Low), the avalanche situation is reported as *no distinct avalanche problem* (EAWS, 2021b).

Lastly, since the winter season 2016-2017, forecasters assign one of three sub-levels to each danger level, except for 1-Low (Techel et al., 2020b, 2022b):

– *plus* or +: the danger is assessed as high within the level, e.g., a $3^+$ is high within 3–Considerable.

– *neutral* or =: the danger is assessed as about in the middle of the level, e.g., a $3^=$ is about in the middle of 3–Considerable.

– *minus* or -: the danger is assessed as low within the level, e.g., a $3^-$ is low within 3–Considerable.

After six years of internal usage, sub-levels are published in the bulletin since the forecasting season 2022-2023 (Lucas et al., 2023; SLF, 2023). Thus, although the bulletin example for 29 January 2021 shown in Fig. 1 displays this sub-level information, it was not published at the time. When referring to sub-levels, we always mean the combination of danger level with sub-level modifier.

## 3 Danger-level model

Using a 20-year historical data set of snow-cover simulations in Switzerland and information about the avalanche conditions in the region, Pérez-Guillén et al. (2022) developed two random-forest models to predict the avalanche danger level for dry-snow

---

[1] We refer to danger levels either using the integer-signal word combination (e.g., 2-Moderate) or by using the integer value only (e.g., 2).



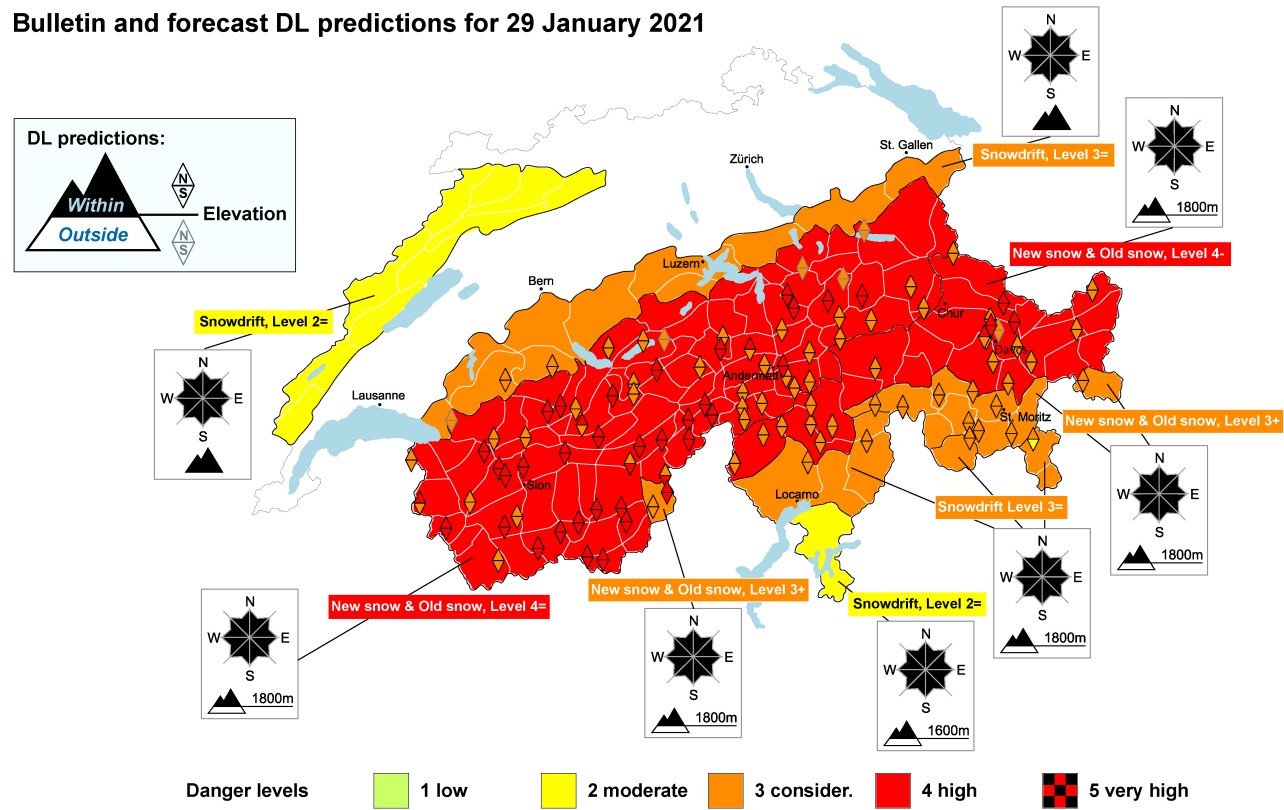

**Figure 1.** Map of Switzerland showing the forecast danger level (background colour) as published in the bulletin issued on 29 January 2021. In addition, the *forecast* predictions of the danger level model are shown for each weather station (represented by triangles). The black contour lines indicate the boundaries of each danger region. Critical elevations and slope aspects are graphically indicated on the map, and the main avalanche problem(s) is specified, together with the sub-level assigned to the danger level in that region: plus (+), neutral (=) and minus (-). The model predictions are represented by upward triangles for the North (N) aspect and downward for the South (S) aspect, with the colour representing the danger level. Black contoured triangles denote predictions *within* the aspects and elevation as specified in the bulletin, whereas grey triangles represent predictions *outside* of this aspect-elevation zone.

conditions in Switzerland at the location of more than 100 automated weather stations located mostly above treeline in the Swiss Alps (indicated by triangles in Fig. 1). One of these models was trained using the forecast danger levels published in the avalanche bulletin as a target variable, while the other used a subset of quality-checked danger level labels. The input features for the models comprise a set of meteorological variables, averaged over a 24-hour time window similar to the avalanche forecast, and features extracted from the simulated snow-cover profiles (12:00 LT). Table A1 shows the variables used in the operational model. While the predictions of the two models are highly correlated, their danger level predictions differ sometimes (Pérez-Guillén et al., 2022). This is caused by the fact that the first model reflects the tendency of the human forecasters to over-forecast (to be on the cautious side), while the second model was trained with a data set reflecting the best



hindcast estimate of the danger level. Forecasters opted for live-testing the second model. Hence, we exclusively analyzed this model in this study.

The model predicts a danger level ranging from 1-Low to 4-High (EAWS, 2021a) since danger level 5-Very High was merged with 4-High when training the model. In our classification setup, the final prediction for a given input $\mathbf{x}$ is the average prediction of each tree $t$ in the forest composed of $T = 1000$ trees:

$$P(D_l|\mathbf{x}) = \frac{1}{T}\sum_{t=1}^{T} p_t(D_l|\mathbf{x}) \tag{1}$$

where $T = 1000$ is the number of estimators in the model, $D_l$ where $l \in \{1,\ldots,4\}$ is the danger level on the scale from 1-Low ($D_1$) to 4-High ($D_4$), and $P(D_l|\mathbf{x})$ is the mean class probability estimate or confidence score representing the estimated likelihood of belonging to each class (danger level). The sum of the probability estimates across the four classes equals 1. Hence, the final predicted danger level, $D(\mathbf{x})$, is the one showing the highest probability $P(D_l|\mathbf{x})$:

$$D(\mathbf{x}) = \underset{D_l}{\mathrm{argmax}}\, P(D_l|\mathbf{x}),\ \text{for } D_l \text{ with } l \in \{1,\ldots,4\} \tag{2}$$

Additionally, the expected value, here referred to as the expected danger value (Maissen et al., 2024), is estimated as the weighted average of all probability scores:

$$E_D(\mathbf{x}) = \sum_{l=1}^{4} (P(D_l|\mathbf{x}) \cdot w_l) \tag{3}$$

where $E_D(\mathbf{x}) \in [1,4]$ is the expected danger value for a given input sample and, $w_1 = 1$, $w_2 = 2$, $w_3 = 3$ and $w_4 = 4$ for each danger level. Since danger levels are defined on a discrete scale, this value can be interpreted as a numerical value that could highlight the model's prediction on a continuous scale. For instance, if 40% of the decision trees predict danger level 3 ($P_{D_3} = 0.4$) while the 60% predicts danger level of 4 ($P_{D_4} = 0.6$), the final danger level prediction is 4-High and the resulting expected danger value is $E_D = 3.6$, but with the added benefit of summarizing the probabilities in a single variable.

### 3.1 Live model deployment

The danger level model was live-tested for the first time during the winter season of 2020-2021. The model provided real-time *nowcast* and *forecast* predictions from the end of December 2020 until May 2021. The *nowcast* setup follows these steps (Fig. 2a):

1. Measurements are transmitted every hour from the AWS to a server at SLF.

2. Using these measurements, the snow-cover is simulated for flat terrain and for four virtual slope orientations (North: N, East: E, South: S, and West: W) with a slope angle of 38° (Morin et al., 2020).

3. The input features required for the model are extracted from the snow-cover simulations every three hours.





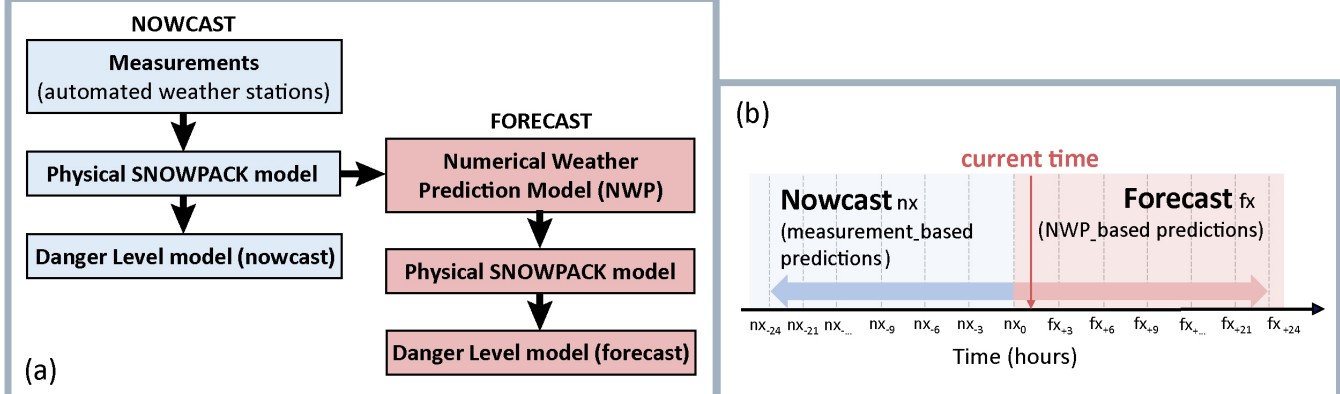

**Figure 2.** Schematic representation of: a) the steps for running the *nowcast* and *forecast* predictions with the danger level model in the live testing deployment, and b) the time intervals applied to resample the input features for the model.

4. Using the extracted input features, the model generates danger level predictions every three hours, referred to as *nowcast* predictions, for each AWS for the flat terrain and the four virtual slope aspects.

In *forecast* mode (Fig. 2a), snow-cover simulations are initiated using the most recent *nowcast* simulations (step 2), and then are driven with COSMO-1 numerical weather prediction model (NWP) data as input (COSMO = Consortium for Small-scale Modeling, https://www.cosmo-model.org/). The NWP data are downscaled to the locations of each AWS to drive snow-cover simulations for the flat study plot and four virtual slope aspects 24 hours ahead. In this case, the meteorological input features of the danger level model are averaged over the next 24 hours of the forecast weather time series to forecast danger levels for each IMIS station and slope aspect. Therefore, for a given time (Fig. 2b), *nowcast* predictions are computed using features based on averaged measurements from the previous 24 hours, while the *forecast* predictions are based on averaged weather predictions for the next 24 hours. For input features that are computed using historical information of more than 24 hours, such as the sum of new snow over three and seven days, the forecast integrates the measurements for the upcoming 24 hours with nowcast values measured from the AWS.

## 4 Data and methods

### 4.1 Test set and model evaluation

For a detailed evaluation of the model's performance, we used the winter season of 2020-2021, during which the model was tested in a live setting for the first time, with only one avalanche forecaster having access to the predictions. We refrain from analyzing later winter seasons, as the model was available to all forecasters from the winter season 2021-2022 onwards, potentially impacting the decision-making process and, hence, the forecast product. During the 2020-2021 season, the model provided predictions on 138 days. On these days, the distribution of danger levels in the public bulletin was imbalanced: 27.4 %



for 1-Low, 38.6 % for 2-Moderate, 31.3 % for 3-Considerable, 2.8 % for 4-High, and 0.04 % for 5-Very High. To assess the overall performance, we first compared the danger level predictions by the model at each station with the danger level forecast in the public bulletin for the corresponding region of the station's location. As the model provides predictions every 3 hours, we filtered the predictions starting at 18:00 (local time) and valid for the next 24 hours to compare with the bulletin, as this is the closest time window to the avalanche bulletin. In addition, we evaluated model performance by considering both station elevation and virtual slope aspects of the input data compared with the critical elevations and aspects specified in the bulletin (Fig. 1). In this context, we defined:

- *Within* predictions: the predictions derived from stations within the core zone of the bulletin, only including stations that were within the elevation range and the active slope aspects mentioned in the bulletin (represented by black contoured triangles in Fig. 1).

- *Outside* predictions: predictions derived from stations not in the elevation range mentioned in the bulletin. In this case, predictions can be further divided into two sub-categories based on whether the slope aspect is active or not in the bulletin (represented by grey contoured triangles Fig. 1).

During the test period, the forecast for dry-snow conditions in the public bulletin indicated that each aspect was active in the following percentages of cases (considering one forecast per region per day): 99.8 % (N), 97.8 % (E), 64.4 % (S), and 95.7 % (W). The prediction data sizes from the model vary per aspect since the model only generates predictions with the condition of a minimum snow-cover depth of 30 cm (Pérez-Guillén et al., 2022), and as the simulated snow-cover depth can differ. Therefore, the predictions are individually analyzed for each slope aspect ($a \in \{F, N, E, S, W\}$) by comparing:

- *nowcast* with *forecast* predictions: $D_{nx,a}$ vs. $D_{fx,a}$

- *forecast* predictions with bulletin forecasts: $D_{fx,a}$ vs. $D_{bu,a}$

- expected danger values computed with the *forecast* predictions with sub-levels forecast in the bulletin: $E_{D,a}$ vs. $D_{bu,a}^{(-=+)}$

For the comparison between expected values and the sub-levels, we applied a simple rounding strategy: *minus* rounds to the nearest numerical danger level minus $\frac{1}{3}$, *neutral* rounds to the integer value of the danger level, and *plus* rounds to $\frac{1}{3}$ above the numerical danger level. The same approach was used to convert the danger level - sub-level combination to a numerical value (Techel et al., 2022a; Maissen et al., 2024). The metrics used to assess model performance are defined in Appendix Sect. B.

## 4.2 Explainability of model

To understand the predictions of machine-learning models, importance values can be derived for each input feature. There are several ways to explore feature attribution. One of these is *SHapley Additive exPlanations* (SHAP) (Lundberg and Lee, 2017), an approach based on integrating concepts from game theory with local explanations. SHAP values provide a quantitative method for assigning importance values either locally (for a single prediction) or across the entire dataset to explain the



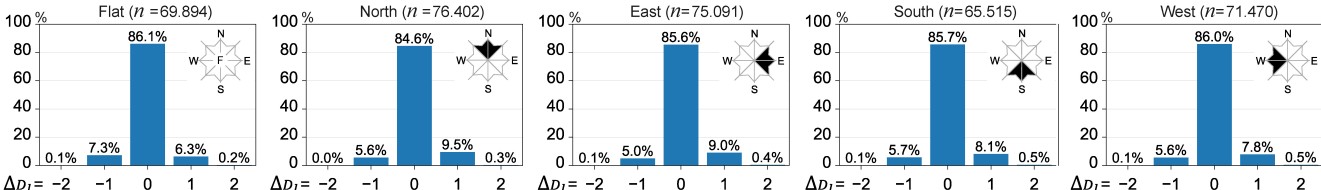

**Figure 3.** Distribution of the differences between *forecast* and *nowcast* predictions ($D_{\mathrm{fx}} - D_{\mathrm{nx}}$) for flat field and the four virtual aspects.

model's overall behaviour (Lundberg et al., 2018). To implement this method, we used the Tree SHAP algorithm designed for tree-based machine learning models, such as random forests (https://shap.readthedocs.io/en/latest/index.html). A detailed description of the computation of the SHAP values is shown in Lundberg et al. (2018). This approach uses an additive feature attribution model, i.e., the output is defined as a linear addition of input features. SHAP values measure the contribution of each input feature, which can be positive or negative with regard to the predicted value. SHAP values are bounded by the output magnitude range of the model, which in our case – a random forest classifier – is from 0 to 1. Therefore, the range of the SHAP values will be from -1 to 1. We applied the SHAP method to obtain global explanations by summarizing the overall importance of features for each class (danger level) of the model, as well as to quantify the contribution of each feature to individual predictions.

## 5 Results

### 5.1 Agreement between *nowcast* and *forecast* predictions

Overall, *forecast* and *nowcast* predictions correlated well, with Spearman correlation $r_s$ ranging from 0.90 for North to 0.91 for Flat ($p < 0.01$). The corresponding agreement rate varied from 84.6 % (North; Fig. 3a) to 86.1 % (Flat; Fig. 3a). When predictions differed, *forecast* predictions were slightly more often one danger level higher than *nowcast* predictions (2-4%), except for Flat (Fig. 3a). Differences by two danger levels were very rare ($\leq 0.5$ %). Comparing the expected danger values ($E_D$, Eq. 3) computed with the model-predicted probabilities showed a high correlation between *forecast* and *nowcast* (Pearson correlation $r_p$ of $0.97 \pm 0.01, p < 0.01$), with a mean difference of $0.0 \pm 0.2$.

### 5.2 Model agreement with avalanche bulletin

#### 5.2.1 Overall agreement

For stations *within* the core zone defined in the bulletin (see Sect. 4.1), the mean agreement rate between the danger level published in the bulletin $D_{\mathrm{bu}}$ and the *forecast* predictions $D_{\mathrm{fx}}$ was 70.4 %, with the lowest agreement rate for the flat-field simulations (66.3 %) and the highest for South aspects (74.0 %; Table 1). South and North predictions generally agreed more with $D_{bu}$ than flat and other aspects. When $D_{\mathrm{bu}} \neq D_{\mathrm{fx}}$, the differences were essentially always by one level, with a larger



**Table 1.** Summary of the model's performance during the operative winter season 2020-2021 based on the *Forecast* predictions: *Within* (all aspects active) or *Outside* (active (+) or non-active (-) slope aspect: Flat (F), North (N), East (E), South (S) and west (W); Support of the predictions; Percentage of samples for each model bias ($\Delta D_{bu,a}$ = value of Eq.B2). Mean absolute errors (MAE) and the Pearson correlation coefficient are computed using the expected danger values and the sub-level assessments.

| Prediction | Aspect | Support | Percentage [%] of $\Delta D_{bu,a}$ = | | | | | $r_p$ | MAE |
|---|---|---|---|---|---|---|---|---|---|
| | | | -2 | -1 | 0 | 1 | 2 | | |
| Forecast *Within* | F | 8665 | 1.3 | 28.2 | 66.3 | 4.2 | 0.0 | 0.88 | 0.37 |
| | N + | 9417 | 0.5 | 16.6 | 72.0 | 10.7 | 0.1 | 0.86 | 0.37 |
| | E + | 9160 | 0.9 | 22.8 | 70.6 | 5.8 | 0.0 | 0.89 | 0.34 |
| | S + | 5737 | 0.9 | 20.6 | 74.0 | 4.6 | 0.0 | 0.93 | 0.31 |
| | W + | 8688 | 1.3 | 25.1 | 68.9 | 4.7 | 0.0 | 0.89 | 0.34 |
| Forecast *Outside* | F | 2152 | 7.3 | 53.8 | 37.0 | 1.9 | 0.0 | 0.74 | 0.48 |
| | N + | 2379 | 4.5 | 39.9 | 51.0 | 4.5 | 0.0 | 0.70 | 0.40 |
| | E + | 2222 | 6.7 | 48.8 | 41.5 | 3.0 | 0.0 | 0.73 | 0.45 |
| | S + | 784 | 4.6 | 37.5 | 54.5 | 3.4 | 0.0 | 0.83 | 0.42 |
| | W + | 1893 | 6.3 | 51.9 | 39.2 | 2.6 | 0.0 | 0.75 | 0.46 |
| Forecast *Outside* | N - | 5 | 20.0 | 60.0 | 20.0 | 0.0 | 0.0 | - | 0.61 |
| | E - | 120 | 0.0 | 98.3 | 1.7 | 0.0 | 0.0 | 0.34 | 0.59 |
| | S - | 1133 | 9.2 | 70.7 | 19.6 | 0.5 | 0.0 | 0.62 | 0.56 |
| | W - | 277 | 18.4 | 69.0 | 12.6 | 0.0 | 0.0 | 0.66 | 0.76 |

share of predictions being one level lower (ranging from 20.6 % (N) to 28.2 % (F) in Table 1) rather than higher (ranging from 4.2 % (F) to 10.7 % (N) in Table 1). In comparably few cases ($\leq 2.0\%$), the difference was by two levels. The agreement rates

were lower for predictions that were partly or fully *outside* the core zone indicated in the bulletin: for instance, for *forecast* predictions (S), they decreased from 74.0% *within* the core zone to 54.5% when below the indicated elevation but within the aspect range, and to 19.6% when fully *outside* the core zone (Table 1).

The issued sub-levels ($D_{bu}^{(-=+)}$; see Section 2) correlated strongly with the *forecast* predictions *within* the core zone ($r_p \geq 0.86, p < 0.01$; Table 1). Considering all aspects, the mean absolute error (MAE) was $\leq 0.37$ *within* the core zone (Table 1),

but was larger for predictions that were partly or fully *outside* the core zone (MAE $\geq 0.4$, MAE $\geq 0.6$, respectively). In other words, predictions partially *outside* the core zone differed by approximately half a level compared to $D_{bu}^{(-=+)}$, which increased to about half to a full danger level fully *outside* the core zone, reflecting the expected differences in avalanche danger as a function of elevation and aspect (Pérez-Guillén et al., 2022; Techel et al., 2022b). Since similar results were observed for the *nowcast* predictions (a detailed overview is shown in Table A2), in the following sections, we analyze exclusively the *forecast*

predictions *within* the core zone.



### 5.2.2 Level-wise agreement

The diagonals of the confusion matrices shown in Fig. 4 represent the proportion (number) of agreements between the forecast of the model and the bulletin ($D_{bu}$; row-wise proportions) for each danger level. These were highest for $D_{bu}$ 1 to 3. The agreement was generally lower for South than for North aspect predictions, which was most evident for levels 2 and 4. In contrast, South predictions showed a considerably higher agreement for $D_{bu} = 1$ than North predictions. Misclassifications (off-diagonal values in Fig. 4) were typically by one level and rarely by two levels (Tables 1 and A2). The row-wise analysis shows that the model often predicted a lower level than the bulletin, particularly for South predictions and for $D_{bu} = 3$ in North predictions. Moreover, danger level 2 was the least precise for both aspects, with a higher number of cases where the model predicted danger level 2 while $D_{bu} = 3$ (Fig. 4).

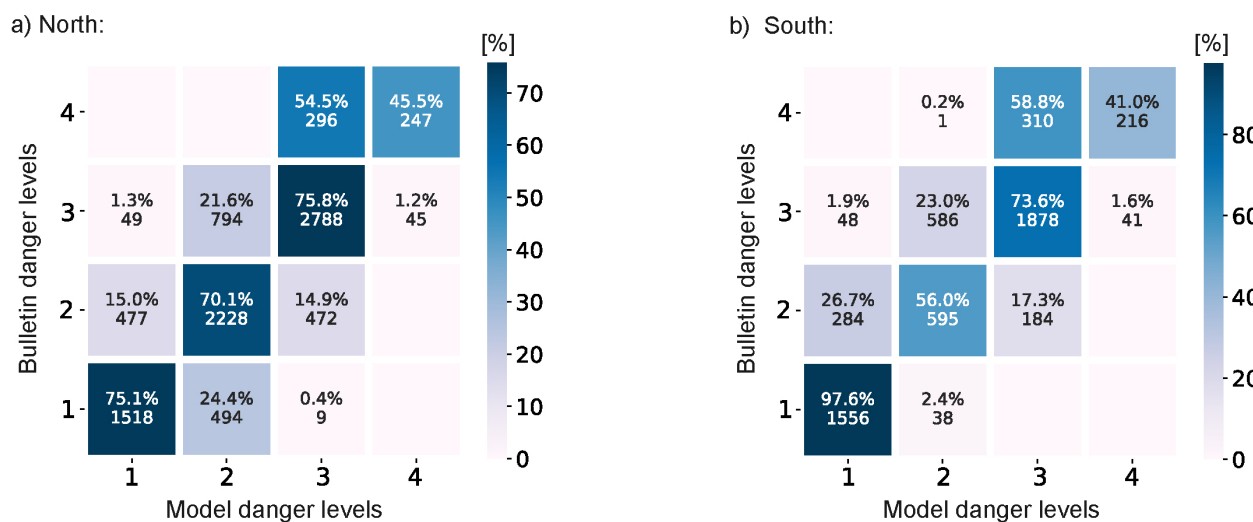

**Figure 4.** Confusion matrices of the *forecast* predictions for the aspects: a) North and b) South. Only predictions from stations *within* the core zone are considered. The colormap shows the percentage of samples per danger level, calculated row-wise.

### 5.2.3 Probability estimates

The majority of model predictions were within one danger level of $D_{bu}$ (Table A2 and Fig. 4). In addition, the distribution of probabilities showed that the model predictions were mainly concentrated between two consecutive levels. Indeed, only a small proportion of the predictions (less than 5 %) estimated probabilities over 0.2 for three levels simultaneously and none of them over 0.1 for all four levels.

Comparisons of the probability distributions (*forecast*-mode, North aspect) with the danger level as forecast in the bulletin are displayed in Fig. 5. We distinguish two cases: when the model and the forecast danger level in the bulletin agreed ($\Delta D_{bu,N} = 0$; Fig. 5a) and when they disagreed (Fig. 5b). When predictions matched (Fig. 5a), the mean probabilities ranged between $P_{D_1} =$



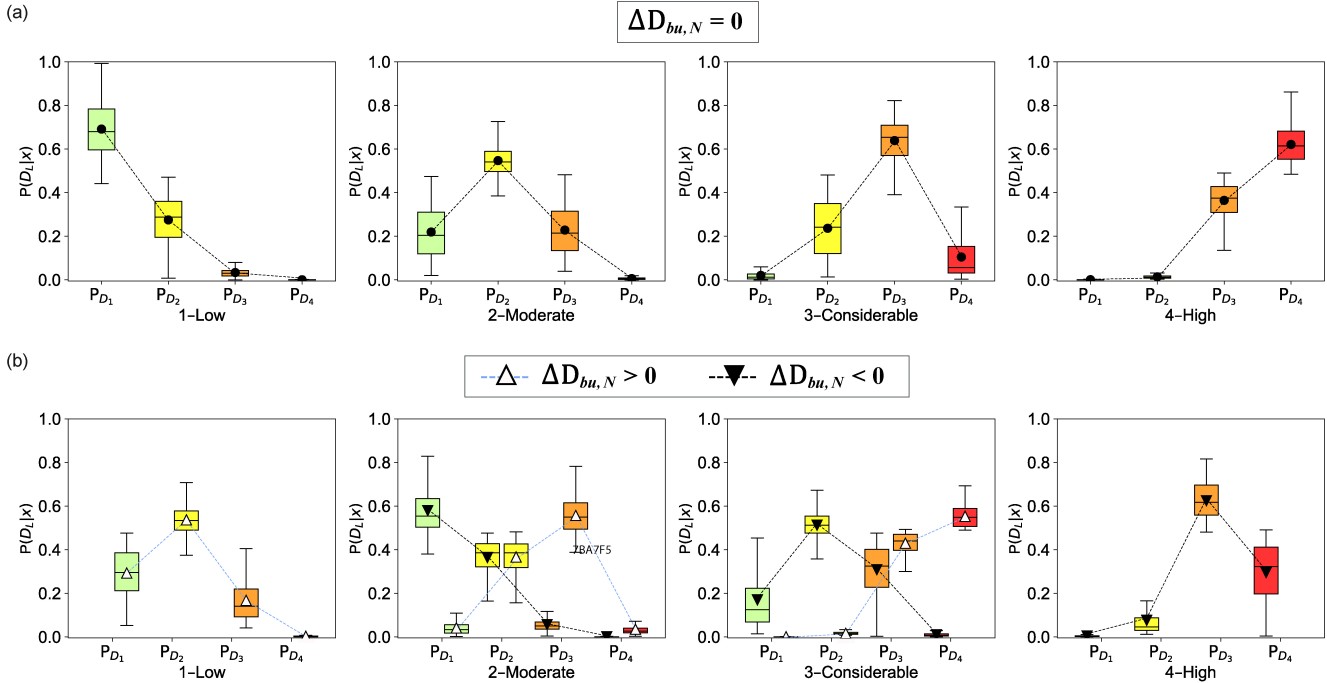

**Figure 5.** Box plot distributions of the probability outputs from the *forecast* predictions for each danger level forecast in the public bulletin split into: a) when the predictions are in agreement with the bulletin ($\Delta D_{bu,N} = 0$) b) when disagreeing: model predictions are higher than the bulletin ($\Delta D_{bu,N} > 0$) or lower ($\Delta D_{bu,N} < 0$). This analysis only considers the *within* predictions for the North aspect (*Within*, N+ predictions of Table 1). The dashed lines link the mean values of the box plot distributions.

0.69 for danger level 1 and $P_{D_2} = 0.55$ for danger level 2. On average, the neighbouring levels had mean probabilities between 0.1 and 0.4. When $D_{bu}$ was 1 or 4, there were rarely any predictions for two levels higher or lower. When the predictions
deviate from the bulletin ($\Delta D_{bu,N} \neq 0$ in Fig. 3b), confidence in predictions decreased, showing higher uncertainty since the mean probability estimates for all danger levels fell below 0.6. Nevertheless, probabilities for the respective level in the bulletin were the second highest (between 0.3 and about 0.45), indicating that the model bias was towards the forecast in the bulletin.

### 5.2.4 Sub-level-wise performance

To compare with the sub-level qualifiers of the avalanche bulletin, we discretized the expected danger values using a simple
rounding strategy rounding to $\frac{1}{3}$, and assigning a *neutral* ($=$) sub-level to its corresponding numerical integer, a sub-level *minus* ($-$) to $\frac{1}{3}$ less than the integer, and similar for a sub-level plus ($+$) being assigned to $\frac{1}{3}$ more than the integer (Sect. 3). The confusion matrix in Fig. B1 displays the agreement rate between the expected danger values computed with the *forecast* predictions and the sub-level qualifiers assigned. Excluding danger level 1, when no sub-level was assigned, the agreement rate between the forecast sub-levels and the expected danger values was, on average, approximately 30 %, ranging from 0.6 %





(4$^=$) to 38.3 %(3$^-$) (diagonal elements in the matrix of Fig. B1). The deviations of the predictions of the model from the main diagonal were mainly concentrated within one sub-level qualifier higher than the bulletin for assessments below 2$^+$ and one sub-level lower for assessments below 4$^-$, showing a decrease in the number of samples with a larger difference. Since the model was trained by merging danger levels 5 and 4, the maximum expected danger values were limited to 4$^=$.

### 5.3 Temporal predictions and avalanche problems

As described in Section 2, avalanche problems provide information about snowpack characteristics in a region. To further investigate the performance of the model across different avalanche situations, we compared the mean agreement rate between model predictions ($D_{fx}$) and bulletin ($D_{bu}$) for each station with the proportion of days that a specific avalanche problem was forecast in this region (Fig. 6). The map in Fig. 6 shows that the proportion of days with a new-snow problem (NS) exceeded 20 % in the Southwestern and central regions of Switzerland, with complimentary patterns for the old-snow problem (OS)

reaching values of >40 % in the central parts of the Swiss Alps (Fig. 6b). Generally, stations in regions with a frequent OS problem showed lower mean agreement rates than those in the central and Northern regions. Consequently, the correlation between the presence of a new-snow problem and the agreement rate was moderately positive ($r_s = 0.42; p < 0.01$; Fig. 6c), while negative for the presence of an old-snow problem ($r_s = -0.52; p < 0.01$, Fig. 6d). The distribution of the wind-slab problem is not shown, as it was forecast for more than 40 % of the days in the Swiss Alps without a clear spatial pattern.

To evaluate the temporal dynamics of the model predictions across avalanche problems, we selected two stations characterized by different avalanche conditions (Fig. 7): Weissfluhjoch (WFJ) located in Davos region in the northeast of Switzerland and Vallée de la Sionne (VDS) in the southwestern. The main differences between these two regions are related to the frequency of snowfall events and the presence of persistent weak layers within the snowpack. In the region of Davos, the NS problem was less frequently forecast compared to Vallée de la Sionne (10% vs 24% of the days) and, conversely, the OS more often (69%

vs 50%). Overall, in both regions, model predictions matched the changes in the danger level as forecast in the bulletin, albeit with different time lags depending on the avalanche situation (Fig. 7). The expected danger values correlated well with the forecast sub-levels (WFJ: $r_p = 0.78$, VDS: $r_p = 0.9$), with the mean absolute error being approximately one sub-level (WFJ: 0.43, VDS: 0.35), showing a better match at VDS than at WFJ. Differences in typical avalanche problems between the regions of Davos and Vallèe de la Sionne might explain the variations in model performance at these two locations. Model predictions

showed a rapid increase in avalanche danger during snow storms (when the NS problem was forecast; Fig. 7). Compared to the bulletin, the model often predicted an increase several hours earlier than the bulletin, driven by the onset of precipitation. While increasing rapidly with precipitation, the expected danger values didn't reach level 4 (Fig. 7) since not all individual decision trees of the random-forest model predicted danger level 4. During times of decreasing avalanche danger, the model predictions often decreased faster than bulletin, particularly for WFJ, as observed in February (Fig. 7) when the expected danger value

decreased from 3 to 1.5. In this situation, the bulletin remained conservative, gradually decreasing from a 3$^+$ at the beginning of the month until reaching 1-Low by the end of the month.



**Figure 6.** Maps showing the proportion of days regionally forecast in the public bulletin during the winter season 2020-2021 for the *New Snow* problem (a) and *Old snow* problem (b). The AWS stations are coloured based on their average agreement rate calculated using the *forecast* predictions (N). Only stations with predictions *within* the core zone on at least 10% of the days are included. Below, scatter plots show the average agreement rate per station against the proportion of days forecast with *New Snow* (c) and *Old snow* (d) problems. Linear regression lines and 95 % confidence intervals are marked in red. The dot sizes are coloured and scaled based on the mean absolute error, computed using the expected danger values and the sub-levels forecast in the bulletin.

## 6 Model explainability

To improve our understanding of the predictions of the model, we employed a Tree explanation model to compute the SHAP values using the test set of the *forecast* predictions (N) during the winter season of 2020-2021. SHAP provides numerical importance values to each feature for every individual prediction of the model (Sec. 4.2).





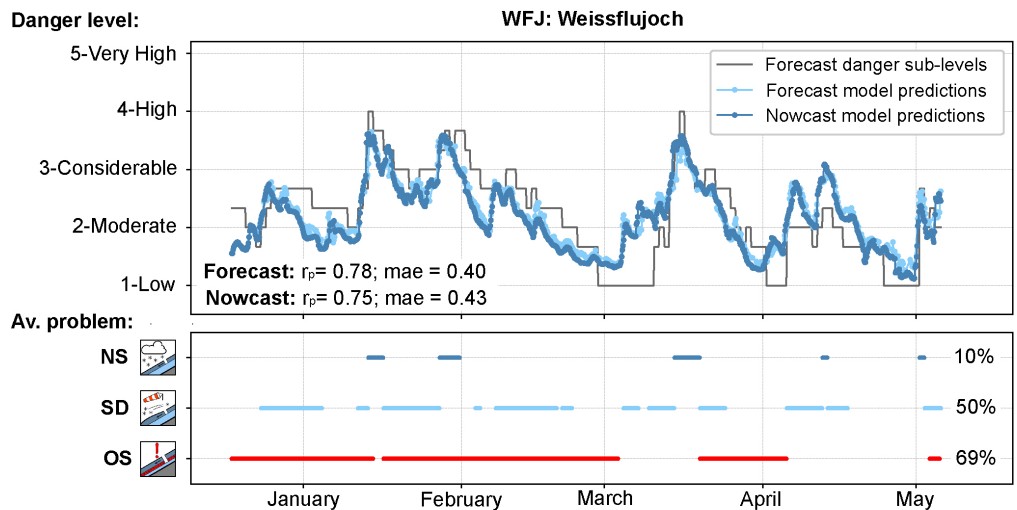

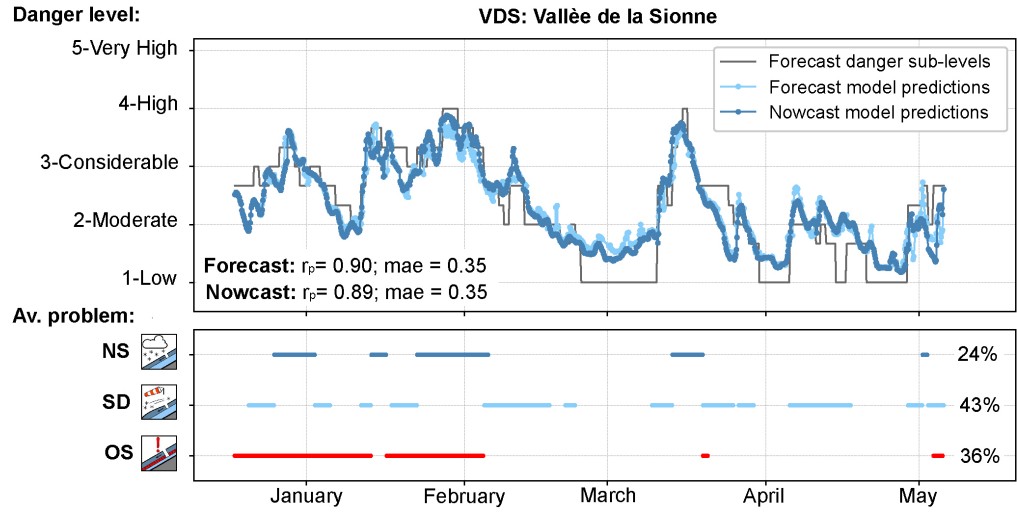

**Figure 7.** Time series of the sub-levels forecast in the bulletin and the expected danger values, computed with the DL *forecast* and *nowcast* predictions from the IMIS stations situated in a) the station of Weissfluhjoch located at 2536 m a.s.l. above Davos at the Eastern Swiss Alps (Fig. 1) and b) the station of Vallèe de la Sionne located at 2385 m a.s.l. above Sion at the western Swiss Alps (Fig. 1). In addition, the forecast avalanche problems for dry-snow conditions during the winter are also displayed: *New-fallen snow* problem (NS), *Wind slab* problem (WS) and *Old snow* problem (OS).

## 6.1 Overall explainability

In general, the 3-day and 7-day sums of new snow (HN24_7d and HN24_3d in Fig. 8) and the skier penetration depth (pen_depth in Fig. 8) rank among the five most important features, regardless of danger level. Nevertheless, the feature importance order and distributions of SHAP values differ for each danger level.





For danger levels 1 and 2, the SHAP beeswarm charts show that high values of variables related to precipitation (e.g. HN24_3d, HN24 and MS_Snow in Fig. 8) and wind or wind drift (e.g. VW, wind_trans24 and wind_trans24_3d in Fig. 8), as well as the skier penetration depth, negatively impact the model's predictions for these classes. The SHAP distributions of these features are fairly symmetric, especially for danger level 1, indicating that low values of these variables impact favourably toward predicting these danger levels. Comparing these danger levels, danger level 1 has distinct SHAP distributions for 7-day

accumulation of new snow and wind drift (HN24_7d and wind_trans24_7d in Fig. 8), with high SHAP values for low amounts of them. Although these features are also important for danger level 2, the distributions show a wider range of SHAP values with different impacts on the model output. Overall, the model tends to predict low danger levels when the input features related to the amount of new snow accumulated by precipitation and wind drift are low. In contrast, for danger levels 3 and 4, the plots show that high values of these features, particularly at danger level 4, have a positive impact (Fig. 8). Precipitation

days are also characterised by high relative humidity, which positively drives predictions towards danger levels 3 and 4. The snow precipitation rate, MS_Snow (Fig. 8), is also an important feature for danger levels higher than 1 (Fig. 8). High rates significantly impact predictions towards danger level 4 and help to discriminate between the other danger levels (Fig. 8), which show an inverse distribution of SHAP values. All these are expected results, as high avalanche hazard situations for dry-snow conditions are mainly characterized by new-snow accumulation during snow storms and the formation of wind slabs due to

strong winds. In addition, low values of input features such as critical cut length and stability indices represent an unstable snowpack, which may also lead to a high avalanche danger. This is also reflected in the SHAP distributions, where low values of these features (e.g. min_ccl_pen, ccl_pwl_100, sn38_pwl_100, S4 in Fig. 8) positively impact high avalanche danger levels (danger levels 3 and 4) and have a negative impact for predicting low avalanche danger levels (danger levels 1 and 2).

    Low danger levels are usually forecast during stable weather conditions, i.e. no precipitation and clear, sunny weather (large

incoming radiation and high fluxes). The SHAP distributions of low danger levels show that low mean values of net long- and short-wave radiation and direct short-wave radiation (LWR_net, Qw, ISWR_dir in Fig. 8) have negative values. Additionally, high mean air temperatures (TA in Fig. 8) positively influenced predictions towards danger level 1, while low temperatures drove model predictions towards danger levels 2 and 3. High values of the variables related to the heat fluxes within the snowpack (Qgo and Qs in Fig. 8) impacted predictions towards lower danger levels, whereas high values of the reflectivity of

the snow surface (pAlbedo in Table A1), which is characteristic of new snow layer, towards high danger levels. Therefore, the model tended to predict low danger levels on days with warm temperatures and stable weather conditions.

## 6.2    Prediction drivers in detail

To gain more insights into the individual impact of the features in the model predictions, we selected an avalanche forecasting period of interest: March 2021 at Weissfluhjoch (WFJ). During this period, $D_{bu}$ increased from level 1 at the beginning of the

month to level 4 on 16 March, followed by a gradual decrease until the end of the month, when it reached level 2 (Figures 7 and 9).

    The time series of SHAP values during this period shows that the accumulation of new snow was the most important factor across all danger levels. For instance, at the beginning and end of the month, low snow accumulations during the last seven and





**Figure 8.** SHAP beeswarm summary of the impact of the 20 most significant features (described in Table A1) on the model output for each danger level: 1-Low, 2-Moderate, Level 3-Considerable and 4-High. These features are sorted by importance for each danger level, showing the most significant variables at the top of each plot. The width of the individual charts indicates the density and frequency of SHAP values, thus providing insights into the relative importance of each feature in relation to the model's output. The colour coding (red for high, blue for low) allows us to understand how changes in a feature's value impact each danger level. While positive SHAP values indicate a change in the expected model prediction towards each model's class, negative values denote the opposite.



last three days (HN24_7d < 14 cm, HN24_3d < 10 cm) resulted in high positive SHAP values for level 1, indicating the highest
probability for this level ($P_{D_1}$ in Fig. 9). Model predictions increased from level 2 to level 3 when new-snow accumulations
exceeded thresholds of 12 cm (HN24_3d) and 20 cm (HN24_7d), approximately. The probability for level 4 was close to zero
during the entire month (Fig. 9), except for the period between March 15 and 17, when $P_{D_4}$ suddenly increased. This increase
was in line with the bulletin. New-snow accumulations of approximately over 17 cm (HN24), 30 cm (HN24_3d), and 40 cm
(HN24_7d) had a positive impact towards predicting danger level 4.

The amount of snow transported by wind (wind_trans24_7d and wind_trans24_3d in Fig. 9) had a similar effect to new
snow but with less impact. Wind drift accumulations over 10 cm in the previous days showed positive SHAP values for high
danger levels and negative values for low danger levels. In addition, high rates of precipitation, MS_Snow > 0.4 (Fig. 9), had
a negative impact for level 1, while the opposite for level 4.

The month was characterized by large skier penetration depths (pen_depth $\geq$ 18 cm in Fig. 9), showing increasingly higher
negative SHAP values for larger depths at danger level 1. When penetration depths exceeded approximately 24 cm and 34 cm,
the SHAP values were high for danger levels 3 and 4, respectively. The interpretation of the minimum critical cut length is
more complex since SHAP values show a variety of ranges for the same feature value, exhibiting both positive and negative
effects, as shown in Fig. 9 for level 2. For higher danger levels, particularly level 4, critical cut lengths below 20 cm showed
positive SHAP values, increasing the probability for this level. The natural stability index was among the seven most important
features for level 2 but with less influence (Sn in Fig. 8). In March, the variation of Sn was minor, showing a low value of Sn
$\leq$ 0.6 (Fig. 9), indicative of an unstable snowpack that favourably influenced this danger level.

A remarkable observation regarding the input values of the meteorological features related to the new snow accumulation
is the bias observed between the HN24, derived from the COSMO weather model for the next 24 hours, and HN24_3d and
HN24_7d (Fig. 9). These features are computed as a combination of extracted values from the snow-cover simulations driven
by COSMO and measured values by the AWS (Sec. 3.1). This explains the differences in the sum of values in Fig. 9 due
to an underestimation of precipitation by the COSMO model from March 15 to 17, which consequently negatively impacted
the *forecast* predictions of the model towards danger level 4. During this period, the expected danger values computed from
*nowcast* predictions were more accurate, approaching danger level 4 (Fig. 6).

## 7 Discussion

### 7.1 Model performance during the operational test

We assessed model predictions using the operational test set during the winter season 2020-2021. This was the first winter
where the model provided predictions in real-time without influencing the avalanche forecasters. Although the model was
trained using data from flat-field snow-cover simulations in *nowcast* mode, it was also tested to predict danger levels for four
slope virtual aspects and in *forecast* mode. Results show that the agreement rates between the *nowcast* and *forecast* danger
level predictions of the model are high, ranging from 84% to 86% (Fig. 3), and the expected danger values are also highly
correlated.



Comparing predictions with the danger level forecasts *within* the core zone of the bulletin, the overall agreement rate varied from 66% (flat) to 74% (South) for *forecast* predictions (Table 1). While the overall agreement rate for flat-field predictions was lower than previously reported (72% in Pérez-Guillén et al., 2022), the agreement rate for flat predictions per danger level was still similar to Pérez-Guillén et al. (2022). The overall decrease is due to the lower support of predictions of danger level 1 in the current study. The performance of predictions *outside* the core zone was consistent with previous studies (Pérez-Guillén et al., 2022; Techel et al., 2022a), showing an increase in the model's bias to predict one danger level lower for all aspects and prediction types (Tables 1 and A2). This bias was more significant when predictions were fully *outside* the core zone.

Comparing the expected danger values of the model with the new sub-level qualifiers of the ordinal danger level scale used operationally by the Swiss avalanche forecasters, deviations in model performance were mainly concentrated within one sub-level qualifier (Fig. B1) and mean absolute errors (MAE) being below 0.4 (see MAE values for *Within* predictions in Tables 1 and A2). Results are consistent for flat-field predictions and aspects as well as for both types of predictions, supporting the operational use of the expected danger values for assessing danger levels in the new sub-level scale (Techel et al., 2022a).

## 7.2 Model Integration for operational avalanche forecasting

Since the winter season 2021-2022, the danger level model has been providing real-time predictions every three hours. The output included a danger level (Eq. 2) as well as an expected danger value (Eq. 3) for each weather station, slope aspect and prediction type (*nowcast* and *forecast*). These predictions were accessible to forecasters, who consulted and compared them with their danger level assessments to issue the daily avalanche bulletin. The model received positive feedback from avalanche forecasters, providing a valuable "second opinion" in the decision-making process (van Herwijnen et al., 2023). Its performance is generally considered comparable to the danger assessments made by forecasters. One of the model's main strengths is its ability to help identify spatial patterns that are usually challenging to assess. This is especially useful when conditions are rapidly changing and evolving, as shown in Fig. 7. A further development aiming to make model predictions more accessible to forecasters is the prediction of a danger level for the predefined warning regions (polygons shown in Fig. 1). This approach, further adapted and live-tested in 2023-2024, generates *forecast* predictions including danger level, sub-level, and corresponding critical aspect- and elevation-information in a format equivalent to human-made forecasts, allowing a largely automatized integration of a "virtual forecaster" in the forecast production process from winter season 2024-2025 onwards (Winkler et al., 2024).

## 7.3 Interpretation of model predictions

Avalanche formation is a complex interplay between the topography, meteorological conditions, and snow-cover properties (Schweizer et al., 2003). Operationally, avalanche forecasting implies predicting the current and future likelihood of avalanches and anticipating their potential size in time and space Schweizer et al. (2020). The time series of the predictions of the model reflected well, and even with a higher temporal resolution, the dynamic nature of avalanche forecasting, showing rapid increases in danger levels correlating with forecasts in the bulletin during or shortly after snowfalls and gradual decreases during periods of no precipitation (Fig. 7). However, the model predictions generally decreased faster than the bulletin when the OS problem





was forecast (Fig. 7). Overall, the model performed better during snow storms, when the NS problem was forecast, and less well in regions characterized by a snowpack with a persistent weak layer (Figures 6 and 7). This suggests that the model predictions are driven mainly by input features related to precipitation rather than variables reflecting the stability of the snowpack.

### 7.3.1 Impact of the meteorological features in the model predictions

Among the main meteorological drivers for dry-snow avalanches is the load from snowfall accumulations during or immediately after storms (Schweizer et al., 2003; Birkeland et al., 2019). Consequently, the amount of new snow is ranked as the most important explanatory meteorological variable linked to avalanche activity in previous studies (Ancey et al., 2004; Hendrikx et al., 2014; Schweizer et al., 2009; Dkengne Sielenou et al., 2021; Mayer et al., 2023). The explainability of the model predictions also showed that the meteorological variables related to accumulated sums of precipitation (HN24, HN24_3d and HN24_7d in Fig. 8) and the skier penetration depth (pen_depth), which is directly linked to new snow parameters, were the most relevant input features of the model for all danger levels, despite variations in the individual rankings (Fig. 8). These features also ranked highest in the internal feature importance scoring by the random forest algorithm when developing the model (Pérez-Guillén et al., 2022). Nevertheless, SHAP values provide a quantitative measure of each feature's impact on the individual model predictions, indicating, as expected, that high accumulations of new snow drive model predictions towards high danger levels, whereas periods of no precipitation or low snow accumulation drive predictions towards low danger levels (Fig. 8). Recent threshold-based binary classification models developed by Mayer et al. (2023) using a subset of avalanche observations in the region of Davos predicted dry-snow avalanche days when new snow accumulations reached thresholds of 22 cm for HN24 and 53 cm for HN24_3d. The detailed explanation of the model predictions at WFJ station (Davos) showed that more than 3 cm and 17 cm of new snow (HN24) had positive SHAP values, increasing the probabilities for predicting danger levels 3 and 4, respectively. Furthermore, a 3-day sum of new snow of 59 cm on 15th March (HN24_3d in Fig. 9) produced a sudden high increase in the probability for danger level 4. Previously developed statistical approaches to predict natural avalanche activity did not include variables for new snow accumulations of more than three days (Dkengne Sielenou et al., 2021; Viallon-Galinier et al.; Mayer et al., 2023). This study shows that HN24_7d ranked between the most important variables impacting all danger levels, especially danger level 1 when there was no precipitation in the previous week at the beginning and end of the month (Fig. 9). Hence, meteorological variables accounting for precipitation over periods longer than three days can also be discriminant features for statistical avalanche forecast models.

Wind is another meteorological factor contributing to slab avalanche formation by transporting snow and producing additional loading in specific areas (Schweizer et al., 2003). In general, the 3-day and 7-day wind-drift accumulations ranked between the 10 most important features for all danger levels, while wind velocity and 1-day wind-drift accumulation were less important (Fig. 8). Particularly, the absence of wind-drift accumulations at the beginning and end of March contributed to model predictions trending towards danger level 1 (Fig. 9). The other main meteorological factors that can promote instability in the snowpack are rapid increases in air temperature and solar radiation due to changes in slab properties (Reuter and Schweizer, 2012). In this regard, the input features used by the model are averaged 24-hour means of temperature and solar radiation (Pérez-Guillén et al., 2022), and therefore do not capture rapid changes, resulting in a secondary impact on model





predictions (Fig. 8). Over the long term, warming and melting processes favour snow metamorphism and settlement, decreasing
the probability of dry-snow avalanche formation (Schweizer et al., 2003). The interpretation of SHAP values is coherent with
this, showing positive SHAP values for air temperatures above 0° (TA in Fig. 9) and generally, for high values of the radiation
parameters for danger level 1 (Fig. 8).

### 7.3.2 Impact of snow-cover features on model predictions

Besides meteorological drivers, snow stratigraphy plays a crucial role in avalanche formation. A prerequisite for dry slab-
415 avalanche release is the existence of a weak layer within the snowpack (Schweizer et al., 2003). The model was developed with
input features extracted from the simulated profiles that contain information on the stability of the snow-cover (Pérez-Guillén
et al., 2022). These features include the critical cut-length computed at two weak layers (min_ccl_pen and ccl_pwl_100 in
Table A1) and several stability indices (Table A1): the natural stability index of two layers (Sn and sn38_pwl_100), the skier
stability index (Ss), and the structural stability index (S4). SHAP feature attribution showed that of these features, min_ccl_pen
had the highest impact on model predictions: low values positively impacted predictions towards danger levels higher than 1
(Fig. 8). Even though the min_ccl_pen-values were low (< 6 cm in Fig. 8) during the period of high avalanche danger forecast
from 15 to 17 March in the region of Davos, the impact was lower than that of precipitation variables for danger levels 3 and 4.
The values of min_ccl_pen remained low until the end of March, while the forecast danger level remained high with a danger
level $3^-$ from 23 to 26 due to OS problem (Fig. 6) and the expected danger value predicted by the model decreased from
425 2.3 to 2.0 (Fig. 9). This highlights the model's weakness in accurately predicting danger levels when the OS problem is the
main contributing factor to avalanche hazard. Additionally, the stability indices did not contribute significantly to high danger
levels in this avalanche situation. Their impact was not particularly high for any danger level (Fig. 8). Mayer et al. (2023) also
tested to develop a classification model using the natural stability index extracted from SNOWPACK simulations to predict
natural dry avalanche activity in Switzerland, resulting in low predictive performance. This was mainly attributed to the limited
discriminatory power of sn38, which aligned with results from previous studies (Jamieson et al., 2007; Reuter et al., 2022).
However, other data-driven models also aimed to predict avalanche activity that included mechanically based stability indices
improved the performance of a classifier that only relied on meteorological and bulk snow parameters simulated with CROCUS
(Jamieson et al., 2007; Reuter et al., 2022).

### 7.3.3 Model explainability and further implementations

SHAP explanations are a powerful approach for quantifying the overall and individual influence of the input features on model
predictions (Lundberg et al., 2018). Compared with other feature scoring methods, such as variable importance based on Gini
importance (Breiman, 2001), SHAP offers a more interpretable analysis of feature contributions. For operational avalanche
forecasting, a representation combining the values of the most important input variables with the SHAP values on a daily
basis, as shown in Fig. 9, can potentially be very valuable for making a complex machine-learning model understandable for
forecasters. This can also be used as a training tool for new forecasters by highlighting relevant parameters and thresholds.
Additionally, this approach could be used to detect outliers in the input features, which may lead to wrong forecasts.



From the perspective of developing new forecasting models based on machine learning, SHAP has confirmed what Pérez-Guillén et al. (2022) qualitatively showed during the model development: the model's predictive performance decreases when the OS problem is the main avalanche problem forecast. This encourages the implementation of new models using better discriminant input variables related to snowpack stability, such as, for instance, recently developed indices assessing snowpack instability (Mayer et al., 2022) or the likelihood of natural avalanches occurring (Mayer et al., 2023). An alternative could be to develop a model to assess danger levels for each avalanche problem. Nevertheless, this approach could be challenging since several avalanche problems were usually simultaneously forecast on the same day and region (Fig. 8).

## 8 Conclusions

In this study, we evaluated the performance of the random-forest classifier predicting the danger level for dry-snow avalanche conditions developed by Pérez-Guillén et al. (2022) during the initial live-testing season in Switzerland. Although the model was trained with a dataset consisting of meteorological measurements providing *nowcast* snow-cover simulations for flat terrain, its performance was equally good, or sometimes even better, in *forecast* mode and when making predictions for four virtual mountain aspects. Overall, the mean agreement rate between model-predicted danger levels and the danger levels as published in the Swiss avalanche forecast was 70%. Moreover, the continuous *expected danger value*, derived using the model-predicted probabilities, showed high correlations ($r_p \geq 0.86$) with the sub-levels as used by the Swiss avalanche warning service. In general, the model reproduced the temporal evolution of avalanche conditions and captured variations in avalanche danger between different slope aspects and elevations, at higher temporal resolution than human-made forecasts. However, the predictive performance was lower during periods characterized by persistent weak layers within the snowpack compared to other situations.

Using the SHAP approach, we showed that model explainability can be greatly increased to obtain a detailed understanding of feature importance overall, but also for individual predictions. We conclude that SHAP could potentially become a powerful tool permitting the interpretation of the predictions of complex 'black-box' machine-learning models, not just in a research context as shown here, but also when using model predictions operationally. Besides increasing the interpretability of such complex models as the danger level model, rigorous validation and implementation of these models remain crucial to ascertain that predictions are reliable, which we consider a prerequisite for operational use. Integrating machine-learning models into avalanche forecasting can complement the forecaster's expertise by providing a "second opinion". Moreover, model predictions may open up possibilities to increase the spatio-temporal resolution of avalanche forecasts (Maissen et al., 2024).

Future developments should include incorporating additional discriminant input features (e.g., indices developed by Mayer et al., 2022, 2023), for instance to increase prediction performance for situations with persistent weak layers, or testing the applicability of these models in regions with different snowpack characteristics.

*Code availability.* The code for the live deployment of the danger level model can be found at deapsnow_live_v1 GitLab repository.



*Data availability.* The data set used to develop the danger level model in this study is available at https://doi.org/10.16904/envidat.330 (Pérez-Guillén et al., 2022) .

*Author contributions.* All authors contributed to the research, with CP leading the analysis and writing. FT, MP, and AV provided support for the analysis, supervision, and proofreading.

*Acknowledgements.* The study to develop and operationally deploy the model was funded by the collaborative data science project scheme of the Swiss Data Science Center (DEAPSnow project: C18-05). We would like to thank all the researchers involved in the DEAPSnow project: Jürg Schweizer, Martin Hendrick, Tasko Olevski, Fernando Pérez-Cruz and Guillaume Obozinski.

*Competing interests.* The authors declare that they have no conflict of interest.





**Figure 9.** Time series of the sub-levels forecast in the bulletin and the expected danger values, computed with the *forecast* predictions from the AWS station situated in Weissflujoch during March, 2021 (up; the starting time for each date was set to 18:00, LT). The probability outputs for each danger level during this period are also displayed. Below, the heatmaps show the overall seven most important features for this forecasting period for each danger level (y-axis) and the time series of the values for each input feature (x-axis). We selected the predictions and input feature values of the time window close to the forecast in the bulletin (starting from 18:00 (LT) on a given day and valid for the next 24 hours). The colour represents the SHAP values computed for each feature in this time window.





# Appendix A: Features of the model and *Nowcast* performance

**Table A1.** Meteorological variables used for training the random forest algorithm. The three types of features are: measured meteorological variable, modelled meteorological variable by SNOWPACK, or extracted variable. Features can be discarded by Recursive Feature Elimination (RFE), manually, or because they are highly correlated with another one.

| Feature description | Feature name |
| --- | --- |
| Mean sensible heat [W/m$^2$] | Qs |
| Mean ground heat at soil interface [W/m$^2$] | Qg0 |
| Mean incoming longwave radiation [W/m$^2$] | ILWR |
| Mean net longwave radiation [W/m$^2$] | LWR_net |
| Mean net shortwave radiation [W/m$^2$] | Qw |
| Mean parametrized albedo [−] | pAlbedo |
| Mean direct incoming shortwave [W/m$^2$] | ISWR_dir |
| Mean air temperature [º] | TA |
| Mean surface temperature [º] | TSS_mod |
| Mean relative humidity [−] | RH |
| Mean wind velocity [m/s] | VW |
| Mean wind velocity drift [m/s] | VW_drift |
| Mean solid precipitation rate [kg/s$^2$/h] | MS_Snow |
| Mean snow height [cm] | HS_mod |
| Mean 24h wind drift [cm] | wind_trans24 |
| Mean 3d wind drift [cm] | wind_trans24_3d |
| Mean 7d wind drift [cm] | wind_trans24_7d |
| Mean 24h height of new snow [cm] | HN24 |
| Mean 3d sum of daily height of new snow [cm] | HN24_3d |
| Mean 7d sum of daily height of new snow [cm] | HN24_7d |
| Mean natural stability index [−] | Sn |
| Mean depth of natural stability index [cm] | zSn |
| Mean Sk38 skier stability index [−] | Ss |
| Mean structural stability index [−] | S4 |
| Natural stability index at surface weak layer [−] | sn38_pwl_100 |
| Critical cut length at surface weak layer [m] | ccl_pwl_100 |
| Min. critical cut length at a deeper layer of the penetration depth [m] | min_ccl_pen |
| Skier penetration depth [cm] | pen_depth |





**Table A2.** Summary of the model's performance during the operative winter season 2020-2021 based on the type of prediction: *Nowcast Within* (all aspects active) or *Outside* (active (+) or non-active (-) slope aspect: Flat (F), North (N), East (E), South (S) and west (W); Support of the predictions; Percentage of samples for each model bias ($\Delta D_{bu,a}$ = value of Eq.B2). Mean absolute errors (MAE) and the Pearson correlation coefficient are computed using the expected danger values and the sub-level assessments.

| Prediction | Aspect | Support | Percentage [%] of $\Delta D_{bu,a}$ = | | | | | $r_p$ | MAE |
|---|---|---|---|---|---|---|---|---|---|
| | | | -2 | -1 | 0 | 1 | 2 | | |
| Nowcast *Within* | F | 8665 | 1.8 | 27.3 | 64.8 | 6.1 | 0.0 | 0.87 | 0.37 |
| | N + | 9417 | 1.0 | 18.4 | 70.2 | 10.2 | 0.1 | 0.85 | 0.37 |
| | E + | 9160 | 1.6 | 25.3 | 66.9 | 6.2 | 0.0 | 0.87 | 0.35 |
| | S + | 5737 | 1.5 | 21.4 | 71.6 | 5.5 | 0.0 | 0.91 | 0.32 |
| | W + | 8688 | 2.0 | 27.0 | 65.4 | 5.7 | 0.0 | 0.87 | 0.36 |
| Nowcast *Outside* | F | 2152 | 8.3 | 52.8 | 36.8 | 2.1 | 0.0 | 0.72 | 0.50 |
| | N + | 2379 | 5.6 | 43.6 | 46.5 | 4.3 | 0.0 | 0.68 | 0.43 |
| | E + | 2222 | 8.1 | 51.2 | 38.2 | 2.5 | 0.0 | 0.71 | 0.48 |
| | S + | 784 | 6.0 | 35.7 | 53.8 | 4.5 | 0.0 | 0.80 | 0.44 |
| | W + | 1893 | 6.9 | 53.1 | 37.1 | 2.8 | 0.0 | 0.73 | 0.48 |
| Nowcast *Outside* | N - | 5 | 20.0 | 80.0 | 0.0 | 0.0 | 0.0 | - | 0.75 |
| | E - | 120 | 1.6 | 97.9 | 0.5 | 0.0 | 0.0 | 0.32 | 0.60 |
| | S - | 1133 | 11.2 | 68.4 | 19.2 | 1.3 | 0.0 | 0.59 | 0.58 |
| | W - | 277 | 26.3 | 63.3 | 10.1 | 0.2 | 0.0 | 0.70 | 0.71 |

## Appendix B: Evaluation metrics

The forecast-nowcast model bias, $\Delta D_{m,a}$, is defined as:

$$\Delta D_{m,a} = D_{fx,a} - D_{nx,a} \tag{B1}$$

where $D_{fx,a}$ is the *forecast*, and $D_{nx,a}$ is the *nowcast* danger level predicted by the model for each mountain slope aspect ($a \in \{F, N, E, S, W\}$). The $\Delta D_{m,a}$ can range from $\pm 3$ for a difference of three danger levels in the scale, $\pm 2$ for two levels, $\pm 1$ for one level, to 0 when the predictions agree. The agreement between predictions is the percentage of total number of cases where $\Delta D_{m,a} = 0$.

The forecast model-bulletin bias, $\Delta D_{bu,a}$, is defined as:

$$\Delta D_{bu,a} = D_{fx,a} - D_{bu,a} \tag{B2}$$

where $D_{fx,a}$ is the *forecast* danger level predicted by the model for each mountain slope aspect ($a \in \{F, N, E, S, W\}$) and $D_{bu,a}$. The $\Delta D_{m,a}$ can range from $\pm 3$ for a difference of three danger levels in the scale, $\pm 2$ for two levels, $\pm 1$ for one level, to 0 when the predictions agree. The agreement between bulletin forecasts and predictions is the percentage of total number of cases where $\Delta D_{bu,a} = 0$.



The Mean Absolute Error (MAE) is the average of the absolute differences between the forecasts and the predicted values:

$$\mathrm{MAE} = \frac{1}{n} \sum_{i=1}^{n} \left| \mathrm{D}_{bu,a}^{(-=+)} - \mathrm{E}_{D,a} \right| \tag{B3}$$

where $n$ is the number of predictions (Support in Tables 1 and A2), $\mathrm{D}_{bu,a}^{(-=+)}$ are the sub-levels forecast in the bulletin and $\mathrm{E}_{D,a}$ is the expected danger values (Eq. 3) predicted for each mountain slope aspect ($a \in \{\mathrm{F, N, E, S, W}\}$).

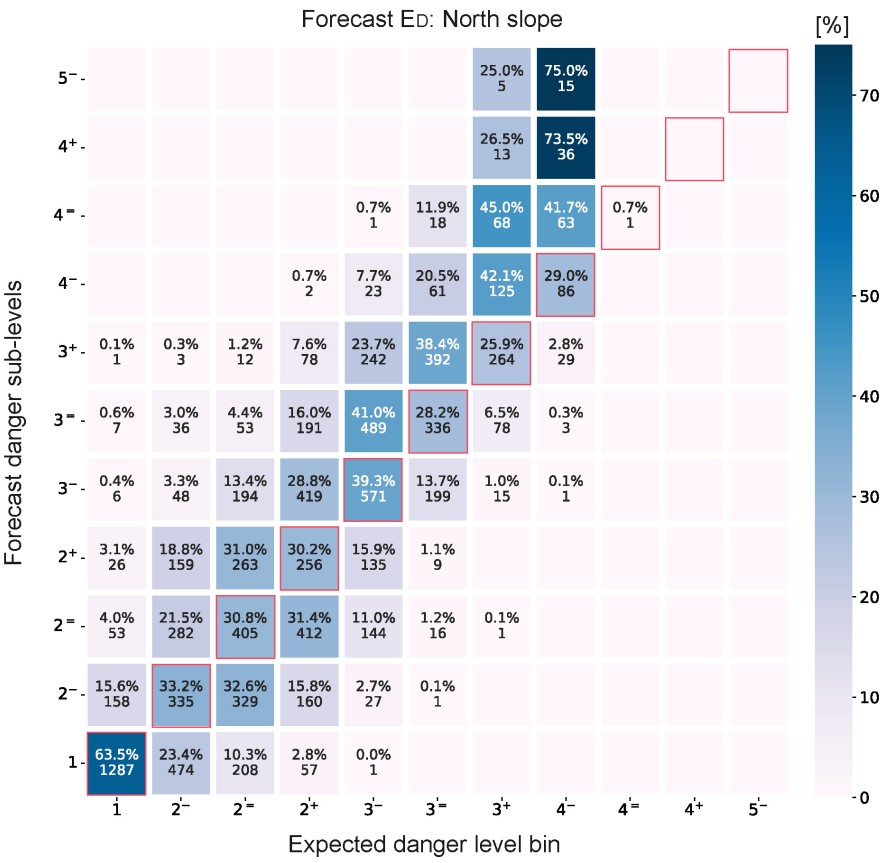

**Figure B1.** Confusion matrix comparing the sub-levels forecast in the bulletin and the expected danger values (proportion of samples: row-wise sum), which are discretized into refined danger level scale bins for the *forecast* predictions.



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
