# Peer review of "Assessing the performance and explainability of an avalanche danger forecast model"

_EGUsphere, 2024_

## Referee Comment (RC3)

**Review EGUsphere 2024-2374**

egusphere-2024-2374.pdf (copernicus.org)
The authors evaluate the performance and interpretability of a random forest model trained to predict avalanche danger levels under dry-snow conditions. Their machine learning model aligns well with human forecaster assessments when used operationally alongside them, providing a higher temporal resolution in danger level assessments. However, the model underperforms in situations where persistent weak layers dictate the avalanche danger.

A challenge in evaluating complex machine learning models, often referred to as "black-box" models, is discerning the rationale behind the model's outputs. The authors employ the SHAP (Shapley Additive Explanations) method to identify and explain which input parameters most strongly influence the model's predictions. By understanding the model's decision-making process, avalanche forecasters can better trust and validate its assessments. This interpretability allows forecasters to compare the model's analysis with their own, offering insight into potential discrepancies and prompting consideration of overlooked factors in their own assessments. Comprehensible model results can also be a valuable tool for forecaster training to showcase which parameters to consider given certain avalanche conditions. This work showcases how to increase the value of machine learning models for operational avalanche forecasting and similar operations.

This study demonstrates how enhancing model interpretability can enhance the practical value of machine learning models in operational avalanche forecasting or similar operations. I recommend to publish this paper after considering my points listed in my detailed comments below.

Kind regards, Karsten Müller.

**Detailed comments**

Paper title: Consider replacing "explainability" by "interpretability".
Line 10: "...though it decreased the performance during periods..." - the performance of what?
Line 111: remove "the" before 60% and "s" in predicts and remove "of" after danger level.
Line 136: "...with only one avalanche forecaster having access to the predictions." Do you mean "only one that is currently on duty" or "only one forecast from the entire forecasting group not necessarily participating in the assessment"? Please clarity.
Table 1: the abbreviation, "delta D_bu,a" is only explained in appendix. Please explain it here if you use it in the table caption.
Line 244: "old snow problem (OS)" - consider replacing the term "old snow" by the EAWS standard name for the avalanche problem "persistent weak layer (PWL)".

Figure 6: Please consider using the same scales in plots a and b and c and d. Or add a note in the caption that different scales are used. "old snow" see comment to line 244.

Line 327 and following paragraph: It is not clear to me what is meant here - please consider rephrasing.

Line 339: Swap words "slope" and "virtual": ...for four virtual slope aspects...

Line 352: Please state more clearly what do you mean by "both types"?

Line 371: put "Schweizer et al., 2020" at all in parenthesis.

Line 373: "rapid increases and danger levels correlating with forecast in the bulletin..." Replace "forecast" with "danger level" or just leave it out.

Line 374: again, consider replacing OS problem by PWL problem.

Line 405: have you considered using "minimum and maximum temperature or temperature difference during the day" or "strongest temperature change over six hours" as an input parameter to capture rapid temperature changes?

Line 431: please check this sentence. Is there a comma missing after stability indices?

Line 577: remove reference to Techel et al. ... discussion.

---

## Author Comment (AC1)

**RC1: Comment on egusphere-2024-2374 (Simon Horton, 14 Aug 2024)**

**General comments**

This paper evaluates a machine learning model that predicts avalanche danger levels in Switzerland. The authors developed the model authors in a previous study, so this study focused on comparing the model's predictions with expert forecasts during the 2020-21 winter across space, time, and under different avalanche conditions. The authors also analyze which model inputs influenced predicted danger in different scenarios, adding transparency and explanations for an otherwise black-box model. The study is well-designed, clearly organized, and effectively communicated, making it both interesting and relevant. Numerical avalanche forecasting is a rapidly developing field with significant implications for public safety and natural hazard management. I recommend publishing this paper in NHESS after addressing the relatively minor comments and clarifications below.

Thank you for your detailed review of our paper. We greatly appreciate your positive feedback and are glad you consider the study well-designed and relevant to the field of numerical avalanche forecasting. Thank you also for highlighting the importance of this research for public safety and natural hazard management. We will address the comments and suggestions you provided in the new version of the manuscript. Thank you for your thoughtful review and for recommending the publication of our work. Please find below the reply to your comments.

Kind regards,

Cristina Pérez-Guillén

**Specific comments**

1. One limitation that could be more clearly emphasized is that the model only accounts for dry snow avalanches. It's important to clarify whether the analysis excluded situations when wet avalanches may have influenced the danger rating, especially since the study period extended into May. According to the EAWS workflow, the danger level is determined based on the highest level indicated by the EAWS matrix for each avalanche problem. If wet snow problems significantly contributed to the danger rating on certain days, those days should be excluded from the evaluation, as the ML model was designed exclusively for dry snow avalanches.

Thank you very much for your suggestion. We will emphasize clearly in the revised manuscript that the model was developed to predict danger levels for dry-snow conditions. You are correct that on some days, the danger level for wet snow conditions was higher than that for dry snow conditions. In Switzerland, two separate danger levels are issued in the public bulletin on those days, corresponding to dry and wet snow conditions. Typically, on such days, the danger levels for dry snow conditions are low (mostly danger level 1 or 2). Thus, we filtered the dataset to include only dry-snow conditions. Since we aim to evaluate the time series of the predictions comprehensively, we chose to use the complete dataset in our evaluation.

2. Within/outside categories. The within/outside categories could be described more clearly. It seems that the "within" group requires both the station elevation to be within the critical elevation range of the bulletin and the virtual slope to be a critical aspect. How are flat slopes within the critical elevation treated? The "outside" predictions are defined as not being in the critical elevation range. But then where would a simulation that falls within the critical elevation range but on a virtual slope that isn't a critical aspect belong? Also, is a subcategory for predictions on critical aspects outside the critical elevation range relevant? It might also be clearer to consistently use "critical" elevations and slope aspects, as in Fig. 1 and Sect. 3, instead of switching to other terms that appear to be synonyms such as "core zone" and "active slope", which may contribute to confusion.

We will clarify and be consistent with the terms used in the revised version of the manuscript. For flat slopes, we grouped the predictions as *Within* the core zone when the station was above the critical elevation and *Outside* otherwise. For virtual aspects, we grouped predictions *Within* when the station's elevation was above the critical elevation and the aspect was considered critical in the bulletin. *Outside* the core zone predictions refer to those below

the elevation range specified in the bulletin. Predictions for virtual aspects are further categorized into those where only one criterion was not fulfilled (i.e., aspect aligns with the forecast but falls below the indicated elevation), and those where neither condition was met.

3. The deviations between the expected danger and sub-levels might stem from the fact that expected values calculated with Eq. 3 would gravitate towards average values and away from extremes. Fig. B1 suggests the sub-level assessments have a wider spread compared to the expected values. This characteristic of expected values could be worth discussing in more detail. (Section 5.2.4 / Fig. B1)

Yes, this is correct. The expected values are constrained between 1 and 4. Additionally, the model rarely predicted danger level 4 with a high probability (> 0.7) (Fig. 5), so the maximum expected values are rarely above 3.5 or within the bin of 4- (Fig. B1). The same applies to the lower limit, danger level 1. However, since no sub-level danger is assigned for this level, the agreement rate is the highest (Fig. B1). We will emphasize this aspect in the revised version.

**Technical comments**

I appreciate several interesting results from this study, including how the model often predicted lower danger than human forecasters, responded to increases/decreases in danger faster than humans, and showed overall poorer performance for persistent weak layer problems. It was encouraging to see the recommendations for improving performance on persistent weak layers, as this seems to be the biggest limitation for operational adoption.

Line 20: Several hundred million Swiss francs "per year"?

It can reach several hundred million Swiss francs, as seen in the catastrophic winter of 1999 [1].

Line 48: Consider a narrative in-text citation to be very clear that "a model" is precisely Pérez-Guillén et al. (2022), which may not be clear with a parenthetical citation.

Thank you very much. We will modify this.

Fig. 1 and 7: These figures label wind-drifted snow problems as "snowdrift/SD" instead of "wind slab/WS" as defined in line 68.

Thank you very much. We will change Figures 1 and 7 according to your suggestion.

Eq. 1: The lowercase probability from each tree (pt) is not defined.

Thank you very much. We defined t in line 98. To clarify further, we will modify this in the new version.

Fig. 2: This is an excellent and clear illustration that helps the reader understand a complex model system.

Thank you for your positive feedback.

Line 157-163: I found these lines difficult to understand until reading Appendix B. Consider moving some details from the appendix to the main text for clarity. This also warrants its own paragraph. I assume that the nowcast versus forecast comparison involved comparing a nowcast with the forecast issued 24 hours earlier — if so, this could be stated explicitly. Additionally, please clarify that the rounding strategy was applied to the expected danger values, as it is initially unclear which variable is being rounded for the comparison.

Thank you for your comment. We will clarify these points in the revised version. Specifically, we will move some definitions from Appendix B to the main text. Additionally, we will explicitly state that the comparison involved a nowcast and a forecast issued 24 hours earlier, ensuring the comparison of the predictions within the same time window. We will also clarify that the rounding strategy was applied to the expected danger values.

Line 183: Were the forecast predictions often higher than nowcast predictions due to a systematic bias in the COSMO

forecasts compared to what was measured at stations? This could be worth discussing later, and perhaps linking with the case where COSMO underestimated precipitation from March 15 to 17.

Based on our analysis, we cannot confirm the existence of a systematic bias in the COSMO forecasts compared to the data measured at the stations. Figure 7 shows that, for predictions at the VDS and WFJ stations, the forecast predictions are not always higher than the nowcast predictions throughout the entire time series. This would be a very interesting topic to investigate in a future study.

Line 192: "Frequently" or "often" are better choices than "essentially always".

Thank you for your comment. We will change it to "mostly".

Fig. 4: Perhaps clarify in the caption that the numbers below the percentages represent counts.

We will modify it.

Sect 5.2.2: It would be interesting to discuss possible reasons for trends in model performance on flat/south/north aspects in the discussion section, perhaps linking with which meteorological/snowpack features may be causing the differences.

Thank you for your suggestion. While this would be very interesting, linking the differences in performance to the variations in snowpack features across different aspects is beyond the scope of this paper.

Line 216: Why is Table A2 with nowcast predictions cited when sections 5.2.2 onward are supposed to focus on forecast predictions? Table 1 seems like a better citation showing many predictions were within one level.

Thank you for your comment. This is an error, and Table 1 should be cited instead. We will correct this in the new version.

Line 227: The phrase "model bias was towards the forecast in the bulletin" is unclear. Bias typically suggests a consistent directional trend, but Fig. 5b shows that when the model assigns the highest probability to a rating different from the bulletin, its second-highest probability often aligns with the bulletin rating. This doesn't necessarily suggest a positive/negative bias.

We will rephrase the sentence to improve clarity.

Fig. 5: The second-row plot for level 2 has some random characters in the middle (7BA7F5).

Thank you very much; we will delete them and update the Figure.

Line 237: The phrase "showing a decrease in the number of samples with a larger difference" is unclear.

We will rephrase this sentence in the new version.

Line 261: The model's response to precipitation several hours earlier might be attributed to its 3-hour temporal resolution, compared to the 24-hour resolution of the bulletin. Similarly, the patterns in Fig. 7 could reflect both the different temporal resolutions and the inherent differences between the two methods. A fairer comparison might involve using only the 1800 LT model predictions. unless the goal was to emphasize the advantages of higher temporal resolution.

To compare the model predictions with the public bulletin forecast and compute the statistics (Tables 1 and A1, Figures 4 and 5), only the model predictions at 1800 LT were used (Lines 142-145). However, in Figure 7, we display the full time series of the model predictions with 3-hour resolution to qualitatively demonstrate the advantage of higher temporal resolution.

Line 288: I agree that the SHAP distribution of MS_Snow for levels 1 and 2 are inverted compared to level 4. however, the distribution for level 3 appears to be scattered.

Yes, you are correct. We will rephrase this sentence in the new version.

Fig. 9: The plot titles for Moderate and Considerable are incorrect. Additionally, could the top panel legend for the black and blue lines use consistent terms from the rest of the paper? I assume the black line is the sub-levels forecast in the bulletin and the blue line is the expected danger from the model.

Thank you very much. We will modify this Figure.

Line 310-311: It would be more intuitive to explain the transition from level 1 to level 2 before discussing the transition from level 2 to level 3. This order would make it easier for readers to follow and avoid confusion (I initially looked at the wrong table when trying to visualize the thresholds). Additionally, it might be helpful to guide readers on how to interpret approximate thresholds from Fig. 9. You could clarify this by stating: "Approximate thresholds for a given danger level can be estimated by identifying feature values when the SHAP values switch from negative to positive" (assuming this was the method).

Thank you very much for your comment. We will improve this in the reviewed paper.

Line 318: MS_Snow is not shown for level 1.

Thank you very much for your comment. This is an error, and Figure 8 should be cited.

Line 324-326: Why would an unstable snowpack with regards to natural avalanches favour level 2? This would make sense for higher danger levels, but I would expect natural avalanches to be unlikely for level 2. The fact low Sn values favour levels 2 and 4, while high Sn values favour levels 1 and 3 suggests the impact of this variable may not be that simple.

We agree with you. As highlighted in previous studies [2, 3, 4], the Sn computed by SNOWPACK shows limited discriminatory ability, and its direct impact by SHAP values is difficult to interpret. We will rephrase this sentence to clarify this point.

Line 333: Why is Fig. 6 cited here?

Thank you very much for your comment. This is an error, and Figure 7 should be cited.

Line 371: Wrong citation style.

Thank you very much for your comment. We will correct this in the revised version.

Appendices: The grouping of tables and figures into 2 appendices seems illogical. Why is Fig. B1 included in an appendix titled "Evaluation Metrics". Consider splitting these into separate appendices.

We will correct this.

Line 490: Dbu,a should be defined here.

We will change it in the new version.

Line 577: Is there a reason both the discussion paper and final paper are listed?

No, we will correct this.

**References**

**1.** Bründl, M. *et al.* Ifkis-a basis for managing avalanche risk in settlements and on roads in switzerland, DOI: 10.5194/nhess-4-257-2004 (2004).

**2.** Jamieson, B., Zeidler, A. & Brown, C. Explanation and limitations of study plot stability indices for forecasting

dry snow slab avalanches in surrounding terrain. *Cold Reg. Sci. Technol.* **50**, 23–34, DOI: 10.1016/j.coldregions. 2007.02.010 (2007).

**3.** Reuter, B. *et al.* Characterizing snow instability with avalanche problem types derived from snow cover simulations. *Cold Reg. Sci. Technol.* **194**, DOI: 10.1016/j.coldregions.2021.103462 (2022).

**4.** Mayer, S., Techel, F., Schweizer, J. & van Herwijnen, A. Prediction of natural dry-snow avalanche activity using physics-based snowpack simulations. *Nat. Hazards Earth Syst. Sci.* **23**, 3445–3465, DOI: 10.5194/ nhess-23-3445-2023 (2023).

---

## Author Comment (AC3)

**RC3: Comment on EGUsphere 2024-2374 (Karsten Müller, 08 Nov 2024)**

The authors evaluate the performance and interpretability of a random forest model trained to predict avalanche danger levels under dry-snow conditions. Their machine learning model aligns well with human forecaster assessments when used operationally alongside them, providing a higher temporal resolution in danger level assessments. However, the model underperforms in situations where persistent weak layers dictate the avalanche danger. A challenge in evaluating complex machine learning models, often referred to as "black- box" models, is discerning the rationale behind the model's outputs. The authors employ the SHAP (Shapley Additive Explanations) method to identify and explain which input parameters most strongly influence the model's predictions. By understanding the model's decision-making process, avalanche forecasters can better trust and validate its assessments. This interpretability allows forecasters to compare the model's analysis with their own, offering insight into potential discrepancies and prompting consideration of overlooked factors in their own assessments. Comprehensible model results can also be a valuable tool for forecaster training to showcase which parameters to consider given certain avalanche conditions. This work showcases how to increase the value of machine learning models for operational avalanche forecasting and similar operations. This study demonstrates how enhancing model interpretability can enhance the practical value of machine learning models in operational avalanche forecasting or similar operations. I recommend to publish this paper after considering my points listed in my detailed comments below. Kind regards, Karsten Müller.

Thank you very much for your review and constructive feedback on our manuscript. We greatly appreciate your positive remarks regarding the potential practical value of our study for avalanche forecasting. We will carefully address all the points outlined in your detailed comments to improve the manuscript. Thank you for your evaluation and recommendation to publish our work. Please find below the reply to your comments.

Kind regards,

Cristina Pérez-Guillén

**Detailed comments**

Paper title: Consider replacing "explainability" by "interpretability".

Thank you very much for your suggestion. However, we think that the term "explainability" is more appropriate in this context.

Line 10: "...though it decreased the performance during periods..." - the performance of what?

We will rephrase the sentence to improve clarity.

Line 111: remove "the" before 60% and "s" in predicts and remove "of" after danger level.

Thank you very much, we will correct this.

Line 136: "...with only one avalanche forecaster having access to the predictions." Do you mean "only one that is currently on duty" or "only one forecast from the entire forecasting group not necessarily participating in the assessment"? Please clarity.

Only Frank Techel from the avalanche warning team forecasters could access the model's predictions. We will rephrase this.

Table 1: the abbreviation, $\Delta D_{bu,a}$ is only explained in appendix. Please explain it here if you use it in the table caption.

We will add the explanation in Tables 1 and B1.

Line 244: "old snow problem (OS)" - consider replacing the term "old snow" by the EAWS standard name for the

avalanche problem "persistent weak layer (PWL)".

Thank you very much for your suggestion. We will replace "old snow" by "persistent weak layer (PWL)" in the new version.

Figure 6: Please consider using the same scales in plots a and b and c and d. Or add a note in the caption that different scales are used. "old snow" see comment to line 244.

We prefer using two scales for each avalanche problem to visualize the differences between regions better. We will add in the caption that different scales are being used.

Line 327 and following paragraph: It is not clear to me what is meant here - please consider rephrasing.

Thank you very much. We will rephrase it for clarity.

Line 339: Swap words "slope" and "virtual": ...for four virtual slope aspects...

Thank you very much. We will do it.

Line 352: Please state more clearly what do you mean by "both types"?

We refer to 'nowcast' and 'forecast' predictions with both types. We will rephrase it.

Line 371: put "Schweizer et al., 2020" at all in parenthesis.

Thank you very much. We will do it.

Line 373: "rapid increases and danger levels correlating with forecast in the bulletin..." Replace "forecast" with "danger level" or just leave it out.

We will do it.

Line 374: again, consider replacing OS problem by PWL problem.

Thank you for the suggestion. We will change it.

Line 405: have you considered using "minimum and maximum temperature or temperature difference during the day" or "strongest temperature change over six hours" as an input parameter to capture rapid temperature changes?

In the initial stages of model development, we tested the model by incorporating maximum, minimum, daily range, and differences for all meteorological features. However, as this did not improve performance, we decided to reduce the number of features and correlations by including only mean values. Nevertheless, features that capture short-term temperature changes could be valuable for testing in future model developments.

Line 431: please check this sentence. Is there a comma missing after stability indices?

We will modify it.

Line 577: remove reference to Techel et al. ... discussion.

We will remove it.

---

## Author Response (AR1)

**Review Egusphere-2024-2374**

Dear Pascal,

We would like to thank you and the three reviewers for your careful review of our paper. We appreciate the time and effort you and the reviewers dedicated to providing valuable feedback on our manuscript. We are grateful for the insightful comments that helped improve the paper. We have incorporated your and the reviewers' suggestions into the revised version of the manuscript. Please find below our responses (in blue) to all the comments and a list of the major changes made to the manuscript. We have also attached the revised version of the manuscript.

Best regards,

Cristina Pérez-Guillén

**List of major changes**

- **Manuscript:** We have revised the entire text based on the review and corrected minor errors.

- **Figures:** We have modified Figures 1, 5, 6, 7 and 9.

- **Tables:** We have updated Tables 1 and B1.

- **Section 4:** We have revised the text based on the review.

- **Discussion:** We have restructured and modified this section based on the review. Some parts have been moved to the Results section.

**Reply to comments from Pascal Haegeli**

1) L48: It seems odd to have a reference at the end of the sentence that describes your research objectives. I assume the reference refers to the model, but this is not necessarily obvious. A possible way for making this clearer could be "... generated by the avalanche danger level model developed by Pérez-Guillén et al. (2022)." I think Simon Horton also commented on this.

Thank you very much. This has been changed in the revised version.

2) L158: Are the (a) subscripts really necessary? I think I understand what they refer to, but the text seems to describe the analysis approach sufficiently already, and these particular subscripts do not seem to be used later in the manuscript. So, they do not seem pertinent.

Thank you very much for your comment. We have modified this part of the manuscript following Simon's review. We would like to keep the subscripts since they are used in Tables 1 and B1, Figure 5, and in the text (for example, in Section 5.2).

3) Some passages in the result section seem to describe methods for the first time (e.g., L239: approach used in Section 5.3), or information about the methods are described in ways that makes it look like they are presented for the first time (e.g., L229 or L268). I always find that a bit confusing: Did I miss something in the methods section?. Please make sure that all details of the analysis approach are described in the methods section (i.e., L164 could be a potential spot for the additional information described later). If you want to describe aspects of the methods again in the result section, word it in a way that makes it clear to the reader that this is just to provide context but not to introduce new information.

Thank you for your comment. Some sentences repeated the methods used as an introduction to the subsection. We have moved some of them to the Methods section and removed them from the Results section in the revised version.

4) L334: Some sections in the discussion seems to introduce substantial amounts of new results (e.g., Section 7.3.2). I think this information should be moved into the results section, which will result in a more focused discussion where the implications of the research can come out more strongly.

We have followed your suggestion and modified the discussion.

5) L354: Section 7.2, which talks about the operational experience with the model, does not directly relate to the study results presented earlier, which makes this subsection different from the preceding and following ones. Hence, it seems to disrupt the discussion of the results. It might therefore be better to move this subsection to the end of the discussion section.

We have moved this section as suggested.

6) L369: The first two sentences of this section seem to be too basic and detailed. A reader who gets to this part of the paper should know this already. If this information is necessary (and cannot be assumed), it could be moved to an earlier part of the manuscript.

We have removed these two sentences.

7) L375: These results are consistent with the results presented in the masters theses of Moses Towell (`https://sfuarp.ca/publications/2019_towell_avprobmodel/` and `https://doi.org/10.5194/nhess-20-3551-2020`) and Heather Hordowick (`https://sfuarp.ca/publications/2022_hordowick_mrm/`). Part of the reason for this pattern might be related to issues in the human assessment dataset: Forecaster hang on to persistent weak layer problems for too long.

Thank you very much for your comment and the references. We have added them in the new version.

**RC1: Comment on egusphere-2024-2374 (Simon Horton, 14 Aug 2024)**

1. One limitation that could be more clearly emphasized is that the model only accounts for dry snow avalanches. It's important to clarify whether the analysis excluded situations when wet avalanches may have influenced the danger rating, especially since the study period extended into May. According to the EAWS workflow, the danger level is determined based on the highest level indicated by the EAWS matrix for each avalanche problem. If wet snow problems significantly contributed to the danger rating on certain days, those days should be excluded from the evaluation, as the ML model was designed exclusively for dry snow avalanches.

Thank you very much for your suggestion. We have now emphasized clearly in the revised manuscript that the model was developed to predict danger levels for dry-snow conditions. You are correct that on some days, the danger level for wet snow conditions was higher than that for dry snow conditions. In Switzerland, two separate danger levels are issued in the public bulletin on those days, corresponding to dry and wet snow conditions. Typically, on such days, the danger levels for dry snow conditions are low (mostly danger level 1 or 2). Thus, we filtered the dataset to include only dry-snow conditions. Since we aim to evaluate the time series of the predictions comprehensively, we chose to use the complete dataset in our evaluation.

2. Within/outside categories. The within/outside categories could be described more clearly. It seems that the "within" group requires both the station elevation to be within the critical elevation range of the bulletin and the virtual slope to be a critical aspect. How are flat slopes within the critical elevation treated? The "outside" predictions are defined as not being in the critical elevation range. But then where would a simulation that falls within the critical elevation range but on a virtual slope that isn't a critical aspect belong? Also, is a subcategory for predictions on critical aspects outside the critical elevation range relevant? It might also be clearer to consistently use "critical" elevations and slope aspects, as in Fig. 1 and Sect. 3, instead of switching to other terms that appear to be synonyms such as "core zone" and "active slope", which may contribute to confusion.

We have clarified to be consistent with the terms used in the revised version of the manuscript. For flat slopes, we grouped the predictions as *Within* the core zone when the station was above the critical elevation and *Outside* otherwise. For virtual aspects, we grouped predictions *Within* when the station's elevation was above the critical elevation and the aspect was considered critical in the bulletin. *Outside* the core zone predictions refer to those below the elevation range specified in the bulletin. Predictions for virtual aspects are further categorized into those where only one criterion was not fulfilled (i.e., aspect aligns with the forecast but falls below the indicated elevation), and those where neither condition was met.

3. The deviations between the expected danger and sub-levels might stem from the fact that expected values calculated with Eq. 3 would gravitate towards average values and away from extremes. Fig. B1 suggests the sub-level assessments have a wider spread compared to the expected values. This characteristic of expected values could be worth discussing in more detail. (Section 5.2.4 / Fig. B1)

Yes, this is correct. The expected values are constrained between 1 and 4. Additionally, the model rarely predicted danger level 4 with a high probability ($> 0.7$) (Fig. 5), so the maximum expected values are rarely above 3.5 or within the bin of 4- (Fig. B1). The same applies to the lower limit, danger level 1. However, since no sub-level danger is assigned for this level, the agreement rate is the highest (Fig. B1). We have emphasized this aspect in the revised version.

**Technical comments**

I appreciate several interesting results from this study, including how the model often predicted lower danger than human forecasters, responded to increases/decreases in danger faster than humans, and showed overall poorer performance for persistent weak layer problems. It was encouraging to see the recommendations for improving performance on persistent weak layers, as this seems to be the biggest limitation for operational adoption.

Line 20: Several hundred million Swiss francs "per year"?

It can reach several hundred million Swiss francs, as seen in the catastrophic winter of 1999 [1].

Line 48: Consider a narrative in-text citation to be very clear that "a model" is precisely Pérez-Guillén et al. (2022), which may not be clear with a parenthetical citation.

Thank you very much. We have modified this.

Fig. 1 and 7: These figures label wind-drifted snow problems as "snowdrift/SD" instead of "wind slab/WS" as defined in line 68.

Thank you very much. We have changed Figures 1 and 7 according to your suggestion.

Eq. 1: The lowercase probability from each tree (pt) is not defined.

Thank you very much. We defined t in line 98. To clarify further, we have modified this in the new version.

Fig. 2: This is an excellent and clear illustration that helps the reader understand a complex model system.

Thank you for your positive feedback.

Line 157-163: I found these lines difficult to understand until reading Appendix B. Consider moving some details from the appendix to the main text for clarity. This also warrants its own paragraph. I assume that the nowcast versus forecast comparison involved comparing a nowcast with the forecast issued 24 hours earlier — if so, this could be stated explicitly. Additionally, please clarify that the rounding strategy was applied to the expected danger values, as it is initially unclear which variable is being rounded for the comparison.

Thank you for your comment. We have clarified these points in the revised version.

Line 183: Were the forecast predictions often higher than nowcast predictions due to a systematic bias in the COSMO forecasts compared to what was measured at stations? This could be worth discussing later, and perhaps linking with the case where COSMO underestimated precipitation from March 15 to 17.

Based on our analysis, we cannot confirm the existence of a systematic bias in the COSMO forecasts compared to the data measured at the stations. Figure 7 shows that, for predictions at the VDS and WFJ stations, the forecast predictions are not always higher than the nowcast predictions throughout the entire time series. This would be a very interesting topic to investigate in a future study.

Line 192: "Frequently" or "often" are better choices than "essentially always".

Thank you for your comment. We have changed it to "mostly".

Fig. 4: Perhaps clarify in the caption that the numbers below the percentages represent counts.

We have modified it.

Sect 5.2.2: It would be interesting to discuss possible reasons for trends in model performance on flat/south/north aspects in the discussion section, perhaps linking with which meteorological/snowpack features may be causing the differences.

Thank you for your suggestion. While this would be very interesting, linking the differences in performance to the variations in snowpack features across different aspects is beyond the scope of this paper.

Line 216: Why is Table A2 with nowcast predictions cited when sections 5.2.2 onward are supposed to focus on forecast predictions? Table 1 seems like a better citation showing many predictions were within one level.

Thank you for your comment. This is an error, and Table 1 should be cited instead. We have corrected this in the

new version.

Line 227: The phrase "model bias was towards the forecast in the bulletin" is unclear. Bias typically suggests a consistent directional trend, but Fig. 5b shows that when the model assigns the highest probability to a rating different from the bulletin, its second-highest probability often aligns with the bulletin rating. This doesn't necessarily suggest a positive/negative bias.

We have rephrased the sentence to improve clarity.

Fig. 5: The second-row plot for level 2 has some random characters in the middle (7BA7F5).

Thank you very much; we have deleted them and updated the Figure.

Line 237: The phrase "showing a decrease in the number of samples with a larger difference" is unclear.

We have rephrased this sentence in the new version.

Line 261: The model's response to precipitation several hours earlier might be attributed to its 3-hour temporal resolution, compared to the 24-hour resolution of the bulletin. Similarly, the patterns in Fig. 7 could reflect both the different temporal resolutions and the inherent differences between the two methods. A fairer comparison might involve using only the 1800 LT model predictions. unless the goal was to emphasize the advantages of higher temporal resolution.

To compare the model predictions with the public bulletin forecast and compute the statistics (Tables 1 and A1, Figures 4 and 5), only the model predictions at 1800 LT were used (Lines 142-145). However, in Figure 7, we display the full time series of the model predictions with 3-hour resolution to qualitatively demonstrate the advantage of higher temporal resolution.

Line 288: I agree that the SHAP distribution of MS_Snow for levels 1 and 2 are inverted compared to level 4. however, the distribution for level 3 appears to be scattered.

Yes, you are correct. We have rephrased this sentence in the new version.

Fig. 9: The plot titles for Moderate and Considerable are incorrect. Additionally, could the top panel legend for the black and blue lines use consistent terms from the rest of the paper? I assume the black line is the sub-levels forecast in the bulletin and the blue line is the expected danger from the model.

Thank you very much. We have modified this Figure.

Line 310-311: It would be more intuitive to explain the transition from level 1 to level 2 before discussing the transition from level 2 to level 3. This order would make it easier for readers to follow and avoid confusion (I initially looked at the wrong table when trying to visualize the thresholds). Additionally, it might be helpful to guide readers on how to interpret approximate thresholds from Fig. 9. You could clarify this by stating: "Approximate thresholds for a given danger level can be estimated by identifying feature values when the SHAP values switch from negative to positive" (assuming this was the method).

Thank you very much for your comment. We have improved this in the reviewed paper.

Line 318: MS_Snow is not shown for level 1.

Thank you very much for your comment. This is an error, and Figure 8 should be cited.

Line 324-326: Why would an unstable snowpack with regards to natural avalanches favour level 2? This would make sense for higher danger levels, but I would expect natural avalanches to be unlikely for level 2. The fact low Sn values favour levels 2 and 4, while high Sn values favour levels 1 and 3 suggests the impact of this variable may not be that simple.

We agree with you. As highlighted in previous studies [2, 3, 4], the Sn computed by SNOWPACK shows limited discriminatory ability, and its direct impact by SHAP values is difficult to interpret. We have rephrased this sentence to clarify this point.

Line 333: Why is Fig. 6 cited here?

Thank you very much for your comment. This is an error, and Figure 7 should be cited.

Line 371: Wrong citation style.

Thank you very much for your comment. We have corrected this in the revised version.

Appendices: The grouping of tables and figures into 2 appendices seems illogical. Why is Fig. B1 included in an appendix titled "Evaluation Metrics". Consider splitting these into separate appendices.

We have corrected this.

Line 490: Dbu,a should be defined here.

We have changed it in the new version.

Line 577: Is there a reason both the discussion paper and final paper are listed?

No, we have corrected this.

**References**

**1.** Bründl, M. *et al.* Ifkis-a basis for managing avalanche risk in settlements and on roads in switzerland, DOI: 10.5194/nhess-4-257-2004 (2004).

**2.** Jamieson, B., Zeidler, A. & Brown, C. Explanation and limitations of study plot stability indices for forecasting dry snow slab avalanches in surrounding terrain. *Cold Reg. Sci. Technol.* **50**, 23–34, DOI: 10.1016/j.coldregions.2007.02.010 (2007).

**3.** Reuter, B. *et al.* Characterizing snow instability with avalanche problem types derived from snow cover simulations. *Cold Reg. Sci. Technol.* **194**, DOI: 10.1016/j.coldregions.2021.103462 (2022).

**4.** Mayer, S., Techel, F., Schweizer, J. & van Herwijnen, A. Prediction of natural dry-snow avalanche activity using physics-based snowpack simulations. *Nat. Hazards Earth Syst. Sci.* **23**, 3445–3465, DOI: 10.5194/nhess-23-3445-2023 (2023).

**RC3: Comment on EGUsphere 2024-2374 (Karsten Müller, 08 Nov 20244)**

Paper title: Consider replacing "explainability" by "interpretability".

Thank you very much for your suggestion. However, we think that the term "explainability" is more appropriate in this context.

Line 10: "...though it decreased the performance during periods..." - the performance of what?

We have rephrased the sentence to improve clarity.

Line 111: remove "the" before 60% and "s" in predicts and remove "of" after danger level.

Thank you very much, we have corrected this.

Line 136: "...with only one avalanche forecaster having access to the predictions." Do you mean "only one that is currently on duty" or "only one forecast from the entire forecasting group not necessarily participating in the assessment"? Please clarity.

Only Frank Techel from the SLF avalanche warning team forecasters could access the model's predictions. We have rephrased this.

Table 1: the abbreviation, $\Delta D_{bu,a}$ is only explained in appendix. Please explain it here if you use it in the table caption.

We have added the explanation in Tables 1 and B1.

Line 244: "old snow problem (OS)" - consider replacing the term "old snow" by the EAWS standard name for the avalanche problem "persistent weak layer (PWL)".

Thank you very much for your suggestion. We have replaced "old snow" by "persistent weak layer (PWL)" in the new version.

Figure 6: Please consider using the same scales in plots a and b and c and d. Or add a note in the caption that different scales are used. "old snow" see comment to line 244.

We prefer using two scales for each avalanche problem to visualize the differences between regions better. We have added in the caption that different scales are being used.

Line 327 and following paragraph: It is not clear to me what is meant here - please consider rephrasing.

Thank you very much. We have rephrased it for clarity.

Line 339: Swap words "slope" and "virtual": ...for four virtual slope aspects...

Thank you very much. We have done it.

Line 352: Please state more clearly what do you mean by "both types"?

We refer to 'nowcast' and 'forecast' predictions with both types. We have rephrased it.

Line 371: put "Schweizer et al., 2020" at all in parenthesis.

Thank you very much. We have done it.

Line 373: "rapid increases and danger levels correlating with forecast in the bulletin..." Replace "forecast" with "danger level" or just leave it out.

We have done it.

Line 374: again, consider replacing OS problem by PWL problem.

Thank you for the suggestion. We have changed it.

Line 405: have you considered using "minimum and maximum temperature or temperature difference during the day" or "strongest temperature change over six hours" as an input parameter to capture rapid temperature changes?

In the initial stages of model development, we tested the model by incorporating maximum, minimum, daily range, and differences for all meteorological features. However, as this did not improve performance, we decided to reduce the number of features and correlations by including only mean values. Nevertheless, features that capture short-term temperature changes could be valuable for testing in future model developments.

Line 431: please check this sentence. Is there a comma missing after stability indices?

We have modified it.

Line 577: remove reference to Techel et al. ... discussion.

We have removed it.